# In situ atomic-scale observation of oxidation and decomposition processes in nanocrystalline alloys

Jinming Guo[1], Georg Haberfehlner[2], Julian Rosalie[1], Lei Li[3], María Jazmin Duarte[4], Gerald Kothleitner[2], Gerhard Dehm [4], Yunbin He [3], Reinhard Pippan[1] & Zaoli Zhang[1,5]

Oxygen contamination is a problem which inevitably occurs during severe plastic deformation of metallic powders by exposure to air. Although this contamination can change the morphology and properties of the consolidated materials, there is a lack of detailed information about the behavior of oxygen in nanocrystalline alloys. In this study, aberration-corrected high-resolution transmission electron microscopy and associated techniques are used to investigate the behavior of oxygen during in situ heating of highly strained Cu–Fe alloys. Contrary to expectations, oxide formation occurs prior to the decomposition of the metastable Cu–Fe solid solution. This oxide formation commences at relatively low temperatures, generating nanosized clusters of firstly CuO and later $Fe_2O_3$. The orientation relationship between these clusters and the matrix differs from that observed in conventional steels. These findings provide a direct observation of oxide formation in single-phase Cu–Fe composites and offer a pathway for the design of nanocrystalline materials strengthened by oxide dispersions.

[1] Erich Schmid Institute of Materials Science, Austrian Academy of Sciences, Leoben 8700, Austria. [2] Institute for Electron Microscopy and Nanoanalysis, Graz University of Technology, Steyrergasse 17, Graz 8010, Austria. [3] Hubei Collaborative Innovation Center for Advanced Organic Chemical Materials, Ministry-of-Education Key Laboratory of Green Preparation and Application for Functional Materials, Hubei Key Lab of Ferro & Piezoelectric Materials and Devices, School of Materials Science and Engineering, Hubei University, Wuhan 430062, China. [4] Max-Planck Institut für Eisenforschung GmbH, Max-Planck-Straße 1, Düsseldorf 40237, Germany. [5] Department of Materials Physics, Montanuniversität Leoben, 8700 Leoben, Austria. Correspondence and requests for materials should be addressed to Y.H. (email: ybhe@hubu.edu.cn) or to Z.Z. (email: zaoli.zhang@oeaw.ac.at)

Nanostructuring and alloying are strategies to obtain enhanced properties for bulk metals[1–5]. Severe plastic deformation (SPD) can effectively generate novel metallic nanocrystalline materials by drastically refining and mechanically alloying normally immiscible composites[6–10]. Now combined with powders processing technique, SPD is extended to produce nanocrystalline alloys with desirable compositions directly from blended powders without any precasting[11], which is a convenient low-cost route in manufacturing applicable bulk materials.

However, gaseous impurities in the raw materials and introduced during SPD generation in the nanostructures pose a challenge which needs to be addressed prior to industrial application as they will influence material properties[12,13]. Oxygen contamination seems to be inevitable during premixing and consolidation for powders processing, as well as in the sequential straining of deformation[14,15]. Moreover, it indeed has been realized that oxygen could be incorporated into the nanocrystalline samples during processing, generating discrepant morphologies[13], mechanical properties[12,15] and thermal stabilities[13]. However, to our knowledge, the exact status of such contaminant inside the materials remains unclear, and how it behaves during annealing has never been studied.

For decades, it was believed that pure metal precipitates of already dissolved elements will form inside grains or at grain boundaries after annealing, and some reports indeed supported this by observing nanosized precipitates of alleged pure metal elements[16]. However, some different types of carbides were detected inside annealed bulk samples, pointing out that contaminants could affect constituents of precipitates and then induce discrepant properties of nanostructured alloys[17]. Nevertheless, the formation processes of heterogeneous precipitates in nanocrystalline alloys were never investigated directly due to the tiny signal intensities of trace amount of such light elements, which are barely detectable by conventional techniques. The current empirical explanations on the effects of light elements are based on phenomenological assumptions[15,18].

It is known that oxides are efficient additives to produce so-called oxides dispersion-strengthened steels/refractory metals which have been strenuously developed for decades due to their high temperature strength, stability and enhanced ductility[19–29]. Therefore, it is reasonable to assume that the potential existence of oxide clusters may affect the properties and microstructures in nanocrystalline materials. Fortunately, the advent of modern in situ high-resolution transmission electron microscopy (HRTEM), combined with image processing technique, enables probing the mechanism behind complicated physicochemical processes at the atomic scale. For example, nowadays novel phase formation[30,31] and transition[32,33], metal-catalyzed process[34], deformation twinning generation[35], irradiation-induced void formation[36] as well as nanocrystal facet development[37] have been captured in real-time observations.

Stimulated by direct atomic-resolution observation, we systematically studied the thermal behaviors of oxygen in high-pressure torsion (HPT) deformed Cu–Fe alloys by means of in situ spherical aberration-corrected HRTEM using a heating holder, via recording atomic-resolved images, diffraction patterns as well as capturing compositional information by electron energy loss spectroscopy (EELS) and energy dispersive X-ray spectroscopy (EDXS), supplemented by X-ray photoelectron spectroscopy (XPS), synchrotron X-ray diffraction (XRD), atom probe tomography (APT) and density-functional theory (DFT) calculations. The results show that the 75 at.%Cu–25 at.%Fe (75Cu–25Fe) alloy has a single face-centered-cubic (fcc) structure at room temperature after severe deformation. When heated, unexpected oxidation processes forming CuO and $Fe_2O_3$ inside grains are detected at relatively low temperature before the fcc

supersaturated solid solution decomposes into Fe-rich and Cu-rich grains. Lattice coherency between oxides and matrix is determined based on different HRTEM images. This work provides, to our knowledge, the first observation of oxygen behavior in nanocrystalline alloys and demonstrates that stable nanosized oxides can easily form inside grains, which may pose a promising route to manipulate mechanical properties by intentionally incorporating light elements before deformation and also assist in quantitatively investigating on the role of oxygen in the grain refinement and intermixing of nanocrystalline alloys.

## Results

**Detection of oxidation and decomposition processes.** The blended powders of 75Cu–25Fe are severely deformed forming a single fcc structure after 100 rotations of high-pressure torsion (equivalent strain of 1360 at the investigated areas). The grains are drastically refined from roughly 50 μm down to an average size of 58 nm according to large population statistics (Supplementary Figure 1 and Note 1). The presence of oxygen in as-deformed bulk samples is verified by XPS measurements and the oxygen concentration is about 3.5 at.% (Supplementary Figure 2, Note 2 and Table 1). The sources of oxygen contamination can be from native oxide layers of the micron-sized particles of raw powders which are usually introduced during powders premixing in air, and residual air voids after the compaction of mixed powders which are trapped inside samples. To reveal the latent oxygen behavior inside HPT nanocrystalline alloys, in situ TEM is employed to observe the subsequent phenomena under annealing. To minimize the possible interference of the surface adsorbed oxygen on the TEM samples, TEM samples are transferred to the microscope immediately after ion-milling. Figure 1 shows low magnification TEM images and diffraction information as well as bright-field (BF) and dark-field (DF) images of the in situ heated sample. Figure 1a displays a series of images from the same area recorded at different temperatures. Almost no grain coarsening is observed even when heated up to 420 °C, and careful investigation with scanning transmission electron microscopy (STEM) confirms the thermal stability of microstructures (Supplementary Figure 3 and Note 3), which has been reported in the literature and attributed to alloying of elements and the pinning effect of contaminants in nanocrystalline metals[38]. The distribution of dark areas in the BF images (Fig. 1a) indicates, that the grain orientations remain unchanged during in situ heating, which makes it feasible to record and compare HRTEM images focusing on a small area (marked as white circle in the first panel of Fig. 1a). The integrated profiles of diffraction patterns of the same area using PASAD[39] are shown in Fig. 1b. All peaks in the profiles are indexed as planes of fcc matrix, Fe precipitates or oxides based on the interplanar spacing. It is worth noting that on the left side of $(111)_{fcc}$ peak, i.e., the larger interplanar spacing values than $(111)_{fcc}$, some tiny peaks corresponding to oxides which are marked as $(101)_{CuO}$, $(112)_{Fe_2O_3}$ and $(110)_{CuO}$ appear when the temperature reaches 260 °C. At an even higher temperature of 340 °C additionally $(200)_{Fe}$ and $(211)_{Fe}$ peaks from Fe can also be clearly observed. Selected area electron diffraction gives structural information of a micron-sized area of the TEM sample, and has higher scattering efficiency than XRD. It is therefore more sensitive to an intensity difference compared to XRD. But it is still relatively insensitive to reflect the onset of oxidation and precipitation at the atomic scale, so the way is to take the advantage of HRTEM (as shown in the following section).

Figure 1c shows the annular dark field (ADF) image and corresponding elemental distributions from electron energy loss spectroscopy (EELS) mapping of the $Cu\_L_{2,3}$, $Fe\_L_{2,3}$ and $O\_K$ edges for the sample after in situ annealing at 420 °C. The colorful

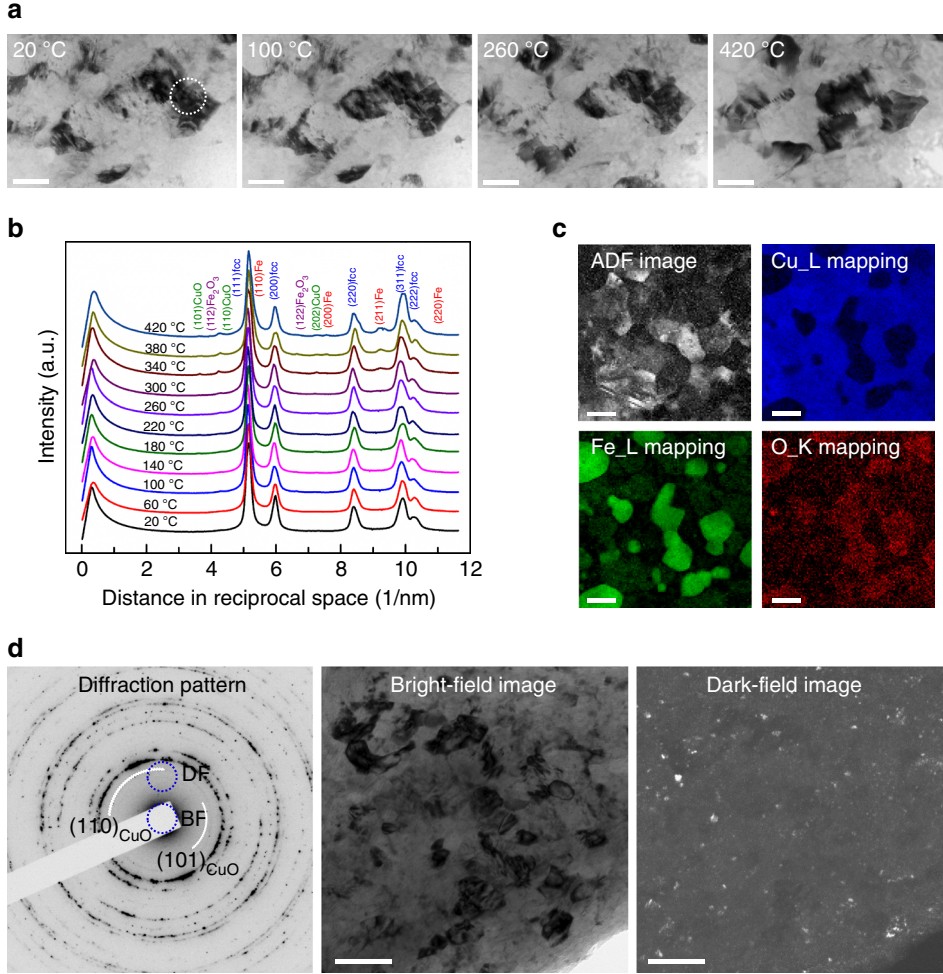

**Fig. 1** TEM images and diffraction patterns for in situ heated samples. **a** Low magnification images focusing on the same area recorded at different temperatures. **b** Integrated profiles of diffraction patterns of the same area. **c** ADF image and corresponding elemental distributions using EELS mapping of $Cu\_L_{2,3}$, $Fe\_L_{2,3}$ and $O\_K$ edges for the sample after in situ annealing at 420 °C. **d** Diffraction pattern of the sample in situ annealed at 420 °C, the corresponding BF and DF images forming from partial diffractions as schematically illustrated in the left diffraction pattern with blue circles. The scale bars in (**a**, **c**) are 50 nm and in (**d**) is 100 nm

area in each image represents the respective elemental distribution. It can be seen that Fe precipitates are embedded inside the Cu matrix with a size ranging from about 5–60 nm. O is distributed over the whole area but enriches at Fe-rich particles. Figure 1d displays a diffraction pattern of the sample in situ annealed at 420 °C and corresponding BF and DF images. The smallest objective aperture is positioned at the locations indicated in the diffraction pattern with blue circles to form BF and DF images. For the DF image only oxide diffraction spots are captured by objective aperture, so in this way the dispersively distributed bright features in DF image correspond exactly to oxides. Although it is impossible to distinguish between CuO and $Fe_2O_3$ due to the unavailability of sufficiently small objective aperture, this oxides mapping verifies that the oxygen exists everywhere inside HPT samples which is consistent with the EELS $O\_K$ edge mapping, and simultaneously it shows that oxides form with extremely fine dimensions of roughly <15 nm[19,20].

**Atomic scale distribution of oxides and Fe precipitates**. To observe the formation processes of oxides at the atomic scale, in situ HRTEM is employed. Figure 2 displays a series of HRTEM images recorded during in situ heating and the corresponding

distribution mappings of oxides and Fe precipitates. The first column shows the sequential in situ HRTEM images recorded during heating. It can be seen intuitively that as the temperature increases, some areas appear blurred and their lattice orientations change compared to the fcc matrix. The HRTEM image recorded at 380 °C is enlarged and inserted at the top-right corner to better distinguish the different structures. The second column shows FFT calculations of the corresponding HRTEM images recorded at different temperatures. The FFT results show that the fcc matrix is on $[011]_{fcc}$ zone axis. For the as-deformed sample prior to heating, the FFT is quite "clean" with only diffraction spots of the fcc matrix along the [011] zone axis. When the sample is heated at 60 °C, several tiny spots (red color) appear which are indexed as reflections of CuO according to the calculations of interplanar spacing. Then after annealing at 100 °C, more diffraction spots appear close to the central spot equivalent to larger spacing, which are indexed as planes of $Fe_2O_3$ phase. CuO shows a monoclinic structure with lattice parameters of $a = 4.684$ Å, $b = 3.423$ Å, $c = 5.129$ Å and $\beta = 99.54°$[40], while $Fe_2O_3$ can be recognized having a hexagonal structure with lattice parameters of $a = 5.035$ Å, $b = 5.035$ Å, $c = 13.752$ Å and $\gamma = 120°$ [41,42]. So the phases of CuO and $Fe_2O_3$ can be easily distinguished by the interplanar spacing of low-index planes, such as (002), (101), and so on. When the temperature reaches 260 °C, the FFT becomes

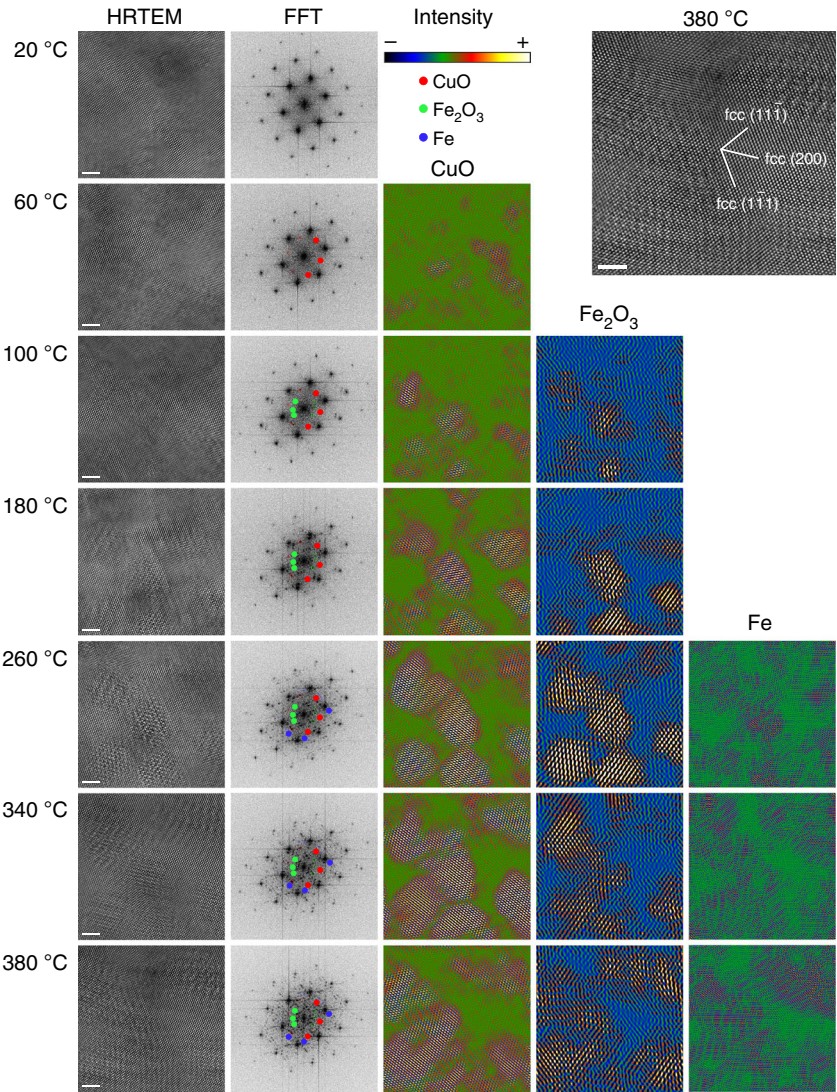

**Fig. 2** HRTEM images and corresponding distributions of oxides and Fe precipitates for in situ heated samples. The first column shows the HRTEM images recorded during in situ sample heating. The second column shows the FFT calculations of the corresponding HRTEM images recorded at different temperatures. The third to fifth columns display the distributions of oxides and Fe precipitates by IFFT. The HRTEM image recorded at 380 °C is enlarged and inserted at the top-right corner. The scale bars in HRTEM images are 2 nm

more complicated with more subtle spots appearing. By carefully comparing the FFT with previous ones, pure Fe precipitates can be identified by their (110) diffraction spots. One should note that, although $(022)_{Fe_2O_3}$ (2.078 Å), $(012)_{CuO}$ (2.034 Å) and $(110)_{Fe}$ (2.037 Å) have similar interplanar spacings, for FFT images obtained at 180 °C and 220 °C, at the circle with a radius of 9.8208 ± 0.4910 1/nm (5% variation) corresponding to interplanar spacing of 2.037 ± 0.102 Å, no clear spots can be detected while other spots from CuO and $Fe_2O_3$ phases can be easily recognized. But for the FFT image of the 260 °C-annealed sample, at this circle region, some discernable spots with detectable intensities show up. So these spots can be justified as the evidence of Fe precipitates. The partial reflections of CuO, $Fe_2O_3$, and Fe (marked as colorized disks) in the FFT images are selected to generate their distribution mappings by Inverse Fast Fourier Transform (IFFT) with specific conditions of the same circle size, circle numbers and positions for each phase[32]. The resultant distributions of CuO, $Fe_2O_3$ and Fe precipitates are displayed in the third to fifth columns respectively, with the same contrast limits for all images in one column. Generally, the areas of the

CuO, $Fe_2O_3$, and Fe precipitates increase with the annealing temperature, and finally the dimensions of the oxides are in the range of 10 nm, well consistent with the result of oxide distributions in DF images shown in Fig. 1d. We note that the spatial distribution of CuO is almost identical to that of $Fe_2O_3$, which means that oxygen diffuses to one specific area during heating, simultaneously triggering the nucleation processes of oxides. From the mapping of Fe precipitates, only tiny areas are marked as Fe agglomerates because of the limited image size.

Additionally, one may query whether the electron beam affects the formation of oxides and precipitates. Actually electron beam effects are unavoidable in TEM studies and should be taken into consideration, especially in in situ experiments[43]. During our experiments, two measures have been taken to minimize the potential influences of the electron beam. First, during image recording, the electron beam was spread to fit the size of the fluorescent screen every time. The beam was switched off during the heating process and the imaging was done immediately after the heating was finished. Second, to confirm that oxidation and precipitation are not caused by e-beam irradiation, a series of

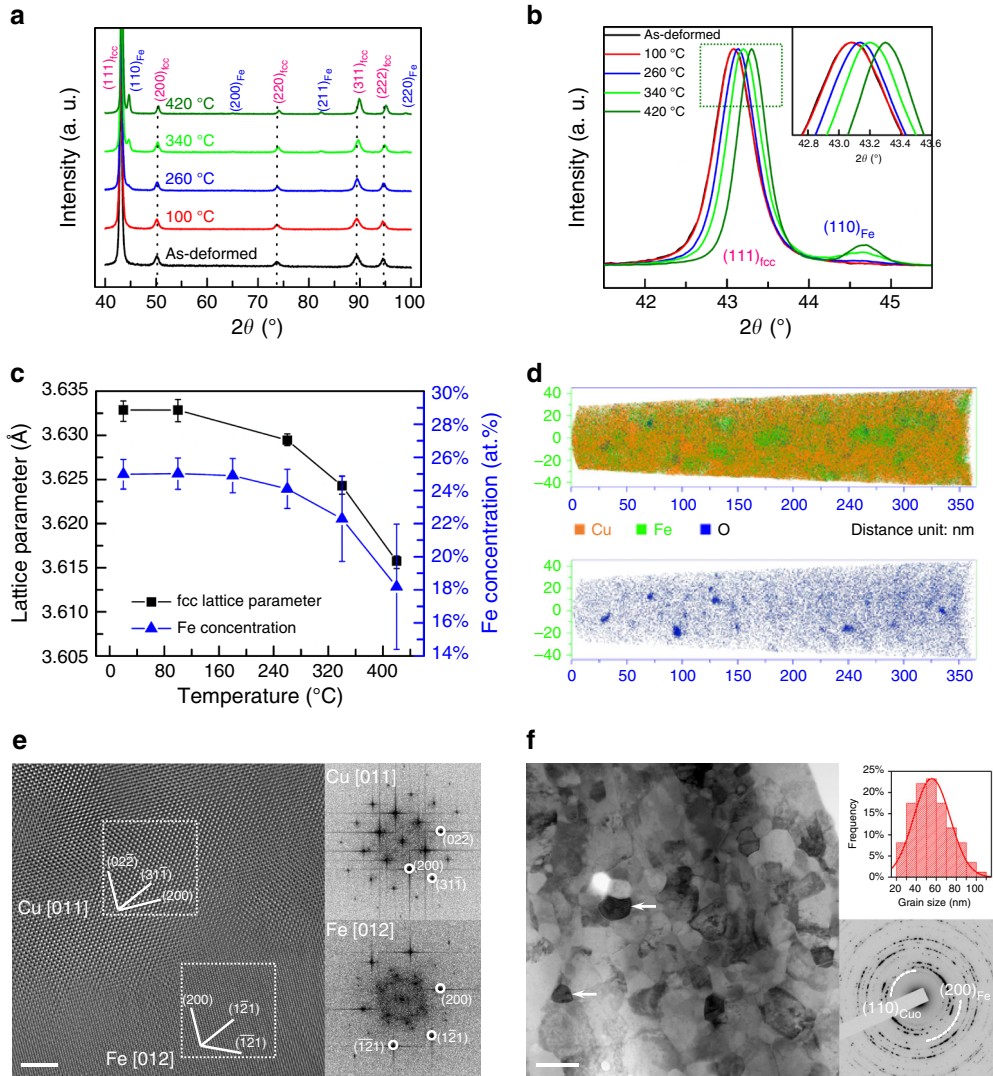

**Fig. 3** XRD, EDXS, APT and TEM characterizations of ex situ annealed samples. **a** Full range XRD profiles of 75Cu–25Fe samples ex situ annealed at different temperatures. **b** Fine scanning XRD profiles focusing on $(111)_{fcc}$ and $(110)_{Fe}$ peaks. **c** Lattice parameter and residual Fe concentration in fcc matrix of samples annealed at different temperatures. **d** APT overview image of the 75Cu–25Fe sample ex situ annealed at 300 °C and the corresponding oxygen map highlighting O-rich clusters. **e** HRTEM image of the sample ex situ annealed at 420 °C showing the neighboring Cu and Fe grains on zone axes of $[011]_{Cu}$ and $[012]_{Fe}$, respectively. The two FFT images are calculated based on the Cu and Fe areas marked with white frames. **f** Low magnification BF image of the sample ex situ annealed at 420 °C; the white arrows indicate Fe grains. The inset at top-right corner is a grain size statistics histogram, and the bottom-right diffraction pattern is from the same annealed sample. The scale bar in (**e**) is 2 nm and in (**f**) is 100 nm

comparison experiments were carried out, where a region of the sample was exposed to the electron beam continuously with the same dose as in the in situ experiments (Supplementary Figure 4 and Note 4). The result shows that even after exposure for several minutes, no obvious changes were observed. In addition, the electron dose rate in the HRTEM images during the in situ heating experiments are quantified as shown in Supplementary Figure 5 and Note 5. Even though electron beam effects are complex, our systematic experiments have confirmed that in the present study electron beam effects on the oxidation and precipitation processes are negligible.

**Oxides and Fe precipitates in ex situ annealed bulk samples**. To investigate whether the TEM sample geometry has an effect on the reactions or processes due to its reduced size, as-deformed bulk materials were ex situ annealed with the same time as in situ heating in above HRTEM investigations. XRD, EDXS, APT and TEM characterizations of ex situ annealed samples are shown in

Fig. 3. Figure 3a shows the full range XRD scan of 40–100° of as-deformed and annealed samples. It shows clearly that the body-centered-cubic (bcc) Fe phase starts to appear when the sample is annealed at 260 °C. Figure 3b displays the finely scanned XRD profiles with a range of 40–47° to closely show the appearing process of $(110)_{Fe}$. It can be observed clearly that, at the position corresponding to $(110)_{Fe}$, a weak peak shows up, which becomes stronger with increasing temperature. Simultaneously, the $(111)_{Cu}$ peak obviously shifts to higher angles as a result of temperature increase above 100 °C. The calculated lattice parameters of the fcc phase, and the averaged residual Fe concentrations in the fcc matrix of samples annealed at different temperatures are presented in Fig. 3c. The lattice parameter of sample annealed at 100 °C keeps identical to that of as-deformed sample, but it then decreases when annealed at 260 °C, and finally it is reduced by 0.017 Å compared to the original value due to the decomposition of super saturated solid solutions after annealing at 420 °C. Based on the two aspects of $(110)_{Fe}$ appearance and $(111)_{Cu}$ peak shift, it

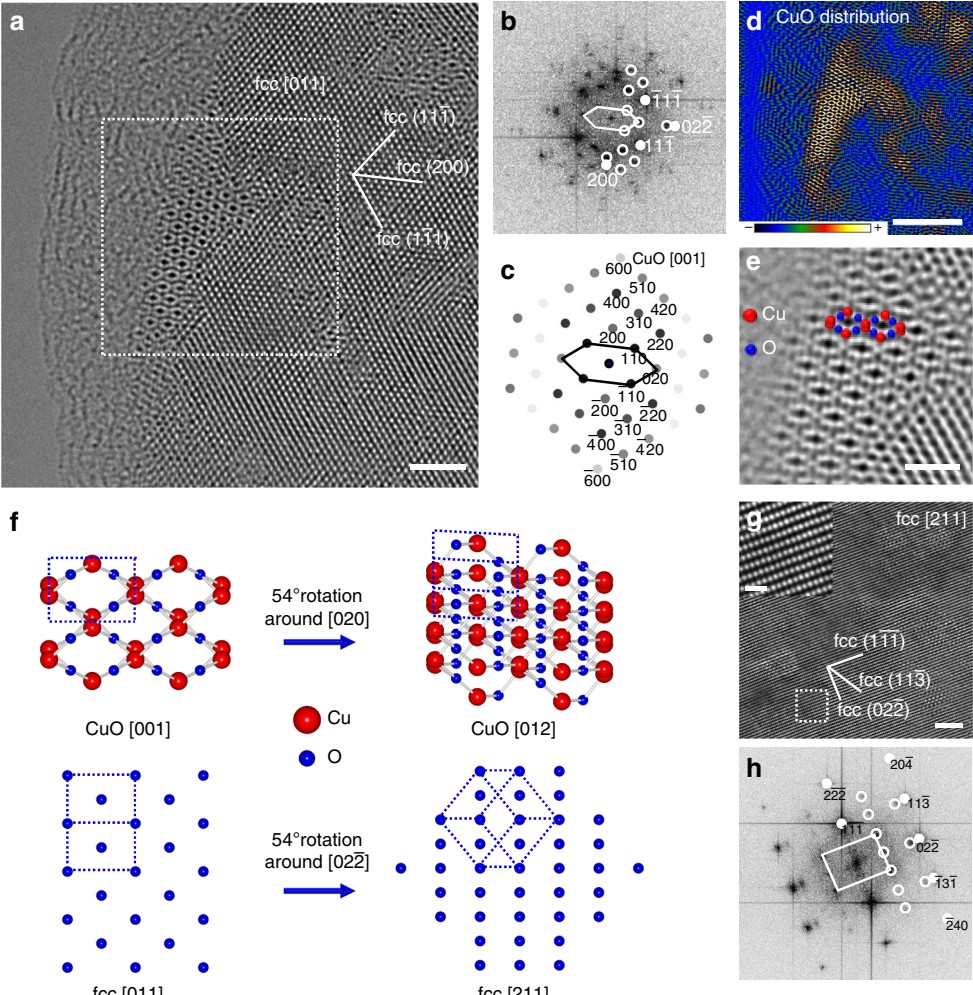

**Fig. 4** Lattice matching relationship between CuO and fcc matrix. **a** HRTEM image of CuO developed from fcc matrix with $[011]_{fcc}$ zone axis. **b** FFT image calculated from the white frame area in **a**. **c** Simulated diffraction pattern of CuO on zone axis of $[001]_{CuO}$. **d** CuO distribution generated from the CuO reflections marked with white circles in **b**. **e** Enlarged filtered CuO HRTEM image. **f** Schematic illustration of lattice matching relationship between CuO and fcc matrix. **g** HRTEM of fcc matrix on zone axis of $[211]_{fcc}$. The inset is an enlarged area of fcc matrix. **h** FFT reflections of image (**g**). Scale bar: (**a**) 2 nm, (**e**) 1 nm, (**g**) 2 nm and the inset in **g** 0.5 nm

can be concluded that the dissolved Fe starts to segregate at the temperature of 260 °C, which is in accordance to in situ heating results. The residual Fe concentration in the fcc matrix is systematically measured at different annealing temperatures using the high spatially resolved EDXS, and the result directly confirms the segregation of Fe from the matrix.

Furthermore, APT was employed to detect the oxide clusters for the ex situ annealed sample at 300 °C as shown in Fig. 3d. From the upper overview image, some Fe-rich areas (green color), containing 8.3 at% of Cu and 0.35 at% oxygen atoms are observed with dimensions of 20–50 nm which accords with the results obtained from EELS mapping displayed in Fig. 1c. The Fe content in the matrix is about 23.3 at% in agreement also with the EDXS result shown in Fig. 3c. The bottom image shows the oxygen distribution, from which it can be seen that O atoms distribute almost homogeneously while some clusters with sizes of 3–8 nm form after ex situ annealing, mainly at the boundaries between Fe-rich grains and the matrix. Figure 3e displays a HRTEM image of the sample ex situ annealed at 420 °C showing neighboring Cu and Fe grains with zone axes of $[011]_{Cu}$ and $[012]_{Fe}$, respectively. The two FFT images are calculated based on the Cu and Fe areas marked by white frames. Evidently, the neighboring Cu and Fe grains have an orientation relationship of $(02\bar{2})_{Cu}//(200)_{Fe}$, which

is different from the typical K–S relationship relevant to fcc and bcc structure interface, i.e., $(111)_{fcc}//(110)_{bcc}$[44]. A low magnification BF image of the same sample ex situ annealed at 420 °C is shown in Fig. 3f. The inset at the top-right corner shows the grain size statistics, and the diffraction pattern at the bottom right is recorded on the same annealed sample where $(110)_{CuO}$ and $(200)_{Fe}$ rings are marked. The average grain size is about 56 nm, and the emergence of $(110)_{CuO}$ diffraction ring confirms the occurrence of oxides inside HPT-deformed samples during annealing. The white arrows in Fig. 3f indicate Fe grains formed after annealing at 420 °C, consistent with the appearance of the $(200)_{Fe}$ diffraction ring.

To confirm the conclusions drawn from the in situ heating experiments, EELS mappings are implemented on the ex situ annealed sample at 420 °C. The morphology and distribution of Fe grains are almost identical to those from in situ annealing, with dimensions of about 20–50 nm (Supplementary Figure 6 and Note 6). Meanwhile, synchrotron XRD measurements presented in Supplementary Figure 7 and Note 7 confirm the presence of oxides inside the materials after annealing.

**Lattice matching relationship between CuO and fcc matrix.** Figure 4a shows a HRTEM image of CuO developing within the

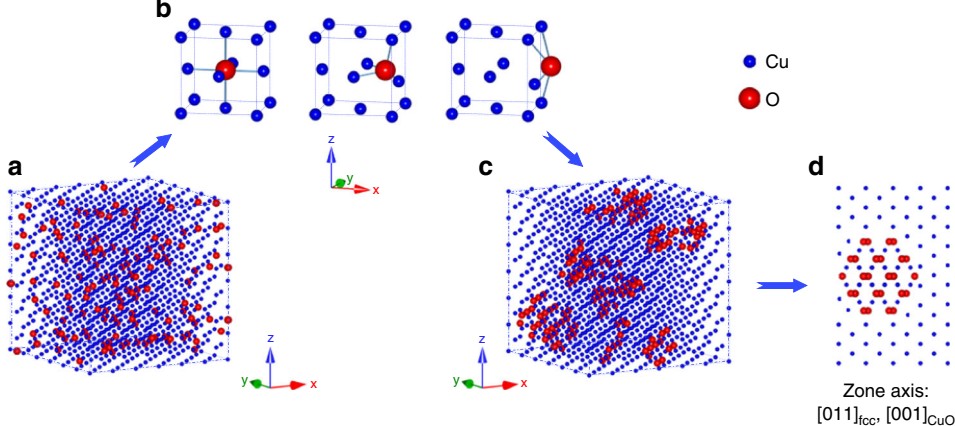

**Fig. 5** Schematic diagram of the oxidation process. **a** As-deformed state with oxygen randomly distributed in fcc matrix. **b** Oxygen movement from one octahedral center to another. **c** O-rich nanoclusters formation inside the fcc matrix. **d** Formed CuO with zone axis of $[001]_{CuO}$ embedded inside fcc matrix with zone axis of $[011]_{fcc}$

fcc matrix after annealing at 420 °C. The fcc matrix shows a lattice structure with a $[011]_{fcc}$ zone axis, while CuO is exactly on the $[001]_{CuO}$ zone axis. The FFT calculated from the area within the white frame in Fig. 4a and simulated diffraction pattern of CuO on $[001]_{CuO}$ zone axis are displayed in Fig. 4b and c respectively. The main reflection spots of fcc matrix on zone axis $[011]_{fcc}$ are marked with solid discs, while the diffraction spots of CuO are marked by white circles, which match well with the simulated diffraction pattern. In the FFT image, it can be clearly seen that the matching relationship between CuO and the fcc matrix is $(\bar{4}00)_{CuO}$ // $(200)_{fcc}$ and $(020)_{CuO}//(02\bar{2})_{fcc}$, with a corresponding lattice mismatch of 6.2% and 10.4% in each direction. Figure 4d shows the IFFT image using only CuO reflections marked in Fig. 4b with white circles highlighting the CuO clusters from fcc matrix as visible in Fig. 4a. To clearly show the CuO lattice structure, an enlarged Wiener-filtered image of CuO is displayed in Fig. 4e, where the Cu and O atoms are identified.

From $[011]_{fcc}$ zone axis of fcc matrix, the matching relationship between CuO and fcc matrix is defined as above-mentioned. Based on crystallography, a 54° rotation of the fcc matrix around the $[02\bar{2}]$ crystallographic orientation to the $[211]_{fcc}$ zone axis, is accompanied by a 54° rotation of CuO around the $[020]$ direction to the $[012]_{CuO}$ zone axis, as shown in Fig. 4f. Figure 4g shows a HRTEM image on a zone axis of $[211]_{fcc}$ with an inset of enlarged lattice structure at the top-left corner. Figure 4h presents the FFT of the whole image shown in Fig. 4g, with the main reflection spots of the fcc matrix on the $[211]_{fcc}$ zone axis marked with white solid discs, and weak spots of CuO on $[012]_{CuO}$ zone axis sorted out by white circles. The matching relationship between CuO and the fcc matrix is accurately defined from two crystallography directions. The full lattice coherency between the oxide nanoclusters and the matrix gives rise to a low interface energy between the two disparate phases, oxide and metal. The low interface energy can effectively prevent the coarsening of the oxide precipitates[19].

## Discussion

Figure 5 shows a schematic diagram of oxidation process at the atomic scale. After severe deformation of the 75Cu–25Fe alloy, the oxygen atoms distribute randomly at the centers of octahedra as interstitials in the fcc lattice as shown in Fig. 5a[45–48]. The oxygen atoms are activated by heating and diffuse in the fcc lattice. The schematic diagram of the activation and diffusion processes of O atoms is shown in Fig. 5b, where the configurations of the initial state, transition state and final state are

displayed in the left, middle and right panels respectively. A number of calculations have reported the energy state of light elements diffusion from one octahedral center to another, giving different activation energy values[45–48]. The activated oxygen atoms then diffuse through the fcc lattice, forming O-rich nanoclusters[47,48] as shown in Fig. 5c. Because of the existence of ample point defects, such as vacancies, solute atoms and O interstitials in severely deformed alloys, the formation energy of O-rich nanoclusters is largely decreased due to the bound state of O-Vacancy pairs with high stability[47]. The binding energy of O with one nearest-neighbor Fe vacancy is about −1.45 eV[47,48]. At the next stage, oxides nucleate when the external heating energy is imposed on the O-rich area. According to the experimental results, the nucleation of oxides starts at 60 °C when the sample is heated for 10 min. Based on the DFT and DFT + U model[49,50], the formation enthalpies of copper and iron oxides are carefully calculated. The calculation procedures and results are displayed in Supplementary Note 8 and Supplementary Table 2. The calculated formation enthalpies are comparable to the values given in many previous reports[51]. It shows that the formation enthalpies of CuO and $Fe_2O_3$ are lower than those of their counterparts, $Cu_2O$ and FeO, respectively. Thus, from the viewpoint of the formation enthalpy, the formation of CuO and $Fe_2O_3$ rather than other oxides with different valences may be explained. The reason why CuO was observed prior to $Fe_2O_3$ is most likely ascribed to the large volume of Cu in the alloy. The finally formed CuO lattice along $[001]_{CuO}$ embedded in the fcc matrix with zone axis of $[011]_{fcc}$ is illustrated in the schematic diagram in Fig. 5d. Nevertheless, it should be emphasized that although nanosized oxide clusters are detected inside grains during heating, the possibility that oxides precipitate at grain boundaries and TEM foil surfaces cannot be excluded.

In summary, our work demonstrates atomic scale observations of oxidation and decomposition processes in severely deformed fcc Cu-based nanocrystalline alloys using in situ HRTEM. Randomly distributed, stable oxide clusters with dimensions of several nanometers are captured in nanocrystalline samples. Our findings show the critical consequences of oxygen presence in nanocrystalline alloys, and may offer a pathway to strengthen the mechanical properties via intentionally forming oxides-dispersed nanocrystalline materials.

## Methods

**Sample deformation by high-pressure torsion**. The commercial coarse-grained powders from Alfa Aesar (Karlsruhe, Germany) of Cu (Purity 99.9%) and Fe (Purity 99.9%) were fully mixed with composition of 75 at% Cu–25 at.% Fe and

then HPT deformed using a two-stage method[11] with 100 rotations at room temperature and air cooling for the whole process. The pressure during HPT was fixed at 7.3 GPa and the rotation speed was 0.4 r/min.

**In situ heating in transmission electron microscope**. In situ heating experiments were conducted in TEM and STEM to characterize in detail the microstructures and compositions of the HPT-deformed Cu–Fe alloy. The TEM sample was in situ annealed for 10 min at different temperatures using Gatan heating holder, in steps of 40 °C from room temperature to a maximum temperature of 420 °C. All microstructural investigations in this work were undertaken at radius of 3.0 mm from the torsional axis of the HPT-deformed disks. (S)TEM studies were carried out using a field emission gun transmission electron microscope (JEOL JEM-2100F, Japan) equipped with an imaging spherical aberration corrector. In STEM mode, a spot size of 0.7 nm of electron beam was used to record high angle annular dark field (HAADF) images. Simultaneous EDXS for nanoscale compositional analysis was also carried out in STEM mode. STEM EELS investigations were done on a probe-corrected FEI Titan³ 60–300 microscope operated at 300 kV equipped with a Gatan GIF Quantum energy filter. The electron beam was perpendicular to the shear plane of the disks for all microstructural investigations in this work.

**X-ray diffraction characterization**. X-ray diffraction was conducted on all samples using a Smartlab X-Ray Diffractometer (Rigaku, Japan) with Cu $K_{\alpha 1}$ radiation ($\lambda = 1.540593$ Å). Here it should be emphasized that the X-ray beam width for all the measurements in this work was limited to 2 mm using the relevant incident slit, covering a large area of HPT-deformed disk from the radius of 2 mm to 4 mm. For the whole range scanning of 40–100°, the scanning speed is 0.4°/min with a step of 0.02°, and then an extremely fine scanning speed of 0.1°/min with the step of 0.02° was imposed for the focused range of 40–47° to get better signal to noise ratio and more accurate diffraction angles.

**Atom probe tomography characterization**. APT measurements were conducted in a local electrode atom probe, LEAP 3000 × HR from Cameca Instruments. The measurements were set up in laser mode with energy of 0.8 nJ and a pulse repetition rate of 250 kHz, base temperature of 60 K and a detection rate of 0.005 atoms per pulse. The specimens were prepared by standard sample lift out and annular milling, using a focused ion beam (FIB) FEI Helios 600i Dual Beam workstation. A final cleaning of the tip with low ion energy of 3 kV ensured a Ga content in the analyzed volumes below 0.05 at.%.

X-ray photoelectron spectroscopy, synchrotron X-ray diffraction methods and density-functional theory calculation procedures are presented in detail in Supplementary Notes 2, 7 and 8.

**Data availability**. The data that support the findings of this study are available from the corresponding authors upon reasonable request.

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

## Acknowledgements

We gratefully acknowledge the financial support by the Austrian Science Fund (FWF): No. P27034 - N20. Peter Kutleša, Gabriele Felber, Herwig Felber and Silke Modritsch at the Erich Schmid Institute of Materials Sciences, Austrian Academy of Sciences, are gratefully acknowledged for their help with the HPT, TEM and metallographic samples preparation. G. H. and G. K. acknowledge funding from the European Union within the 7th Framework Program [FP7/2007–2013] under Grant Agreement No. 312483 (ESTEEM2). Y.H. acknowledges the National Natural Science Foundation of China (Grant Nos. 61274010, 51572073, 11774082). We are highly grateful to Yong Zhang at Hunan University, China and Erich Schmid Institute of Materials Sciences, Austrian Academy of Sciences for his kind help in the DFT calculations.

## Author contributions

Z.Z. conceived the idea and oversaw the whole project. J.G. conducted the experiments and wrote the manuscript with input from all authors. G.H. and G.K. conducted STEM EELS mapping experiments. J.R. assisted the XRD experiments, and L.L., Y.H. performed XPS measurements. M.J.D. and G.D. carried out the APT characterization. J.G. and Z.Z. analyzed the experimental results. R.P. made helpful comments on the manuscript. All authors read through the manuscript and contributed to the discussion of the results.

## Additional information

**Competing interests:** The authors declare no competing interests.

