## [Peer Review File · Nature Communications]

Reviewers' Comments:

Reviewer #1:

Remarks to the Author:

The manuscript addresses the effect of oxygen contamination on nanocrystalline Cu-Fe alloys during annealing using in situ high resolution transmission electron microscopy (TEM) in combination with complementary analysis techniques. The study is very interesting for the field of powder processing materials with important results about the stabilizing effect of small amounts of oxygen on the microstructure during thermal anneal. However, the issues described below need to be addressed before consideration for publication in Nature Communications.

There is no information about how the TEM specimens were prepared. Was there an altered surface layer of the TEM specimens due to ion beam thinning? How thick was the damaged layer? How thick were the TEM specimens in the areas that were studied? Did the TEM specimen thickness correspond to the grain size of the materials?

The authors claim that oxides are formed in the interior of the grains. However, the TEM images are projections of the TEM samples. The presented data do not exclude the possibility of the oxides forming at the TEM specimen surface. Further information is provided before the author can claim that the oxides are formed in the interior of the grains. This is crucial for the statements in the discussion and abstract.

The morphology of the oxides are not shown in the ex situ annealed samples. Why?

What was the partial oxygen pressure in the TEM during observation?

Did the authors determine the oxygen content of the initial raw powders? Did it correlate to the oxygen content in the HPT materials?

Figure 3 in the supplemental material illustrates the presence of Fe grains at 420°. It is not clear how the authors reach the conclusion that the two grains are pure iron grains. Please clarify.

Figure 5 in the supplemental material shows concentration profiles after thermal anneal. The presented data do not verify that the formation of iron occurs as a result of the thermal anneal since there are no data provided before thermal anneal. A figure corresponding to Figure 3 c, in the main manuscript, with data from EELS mapping at 20 °C for both in situ and ex situ would provide crucial information for this manuscript.

There are small areas with bright contrast at the grain boundaries in Figure 5 a in the supplementary material as well as the areas in the white circles. What is the difference between the small bright areas?

Reviewer #2:

Remarks to the Author:

In this manuscript oxidation in nanostructured FeCu alloys is studied using a wide array of experimental techniques and some electronic structure calculations are presented as support as well. It is an interesting manuscript for sure, but I do not deem this of high enough novelty nor of sufficient interest for the readership of Nature Communications. I therefore recommend rejection of the manuscript but I fully endorse the authors submitting this work to a more specialized journal.

For this reason, I provide some constructive feedback to the manuscript that the authors could take into account for future submissions:

* The authors have missed out on a large part of the rather rich literature of oxidation and oxidation kinetics in oxide dispersion strengthened (ODS) steels. These are mentioned very briefly in the introduction but no reference is made to significant studies of kinetics, phase formation, effect of ion beams (or electron beams) on the evolution of these nanostructured alloys.

* The authors give too little information on the impurity levels and characteristics in the as-received materials. It is not trivial to ascertain if low-T oxidation is abnormal or curious, as the authors claim, when we have no clear information on the impurity contents, other than the balance numbers. This information should be available to the authors and will be very useful when drawing conclusions.

* The language of the supplementary materials has to be improved significantly. There is a large discrepancy between the manuscript and the suppl.mater. There are also some small issues with the manuscript itself, but these are minor and should be caught by a third party that proofs it.

* The very weak O 1s signal in the suppl.mater. fig 2 providing an estimate on the composition is far from solid. No error bar or uncertainty is presented either, which is here rather glaring.

* Vanilla DFT applied on transition metal oxides is a well-known problem area. Strong correlation effects can be quite important. The use of the enthalpies here thus calculated is not so significant in the current manuscript but still, one should do this properly or not at all.

* The study in the suppl.mater motivating the claim in the main manuscript that the e-beam has no effect is far from conclusive. I would recommend the authors to improve the quantitative analysis of this effect (or lack thereof).

Reviewer #3:

Remarks to the Author:

This is an interesting article and is worthy of publication. Whether it is novel enough for publication in Nature Communications I will leave to the editor. However, there are a number of issues which I believe need to be addressed before publication:

1. The oxygen analysis by XPS is not very convincing. I would be much more convinced if a piece of pure copper had been ion cleaned and analysed at the same time. Why was a quantitative bulk analysis technique not used to determine the oxygen content of the samples?

2. What is the evidence that the oxide particles are growing within the sample during the in-situ heating and not on the surface of the sample. Indeed the micrograph shown in Fig. 4a looks like a section through the edge of a thin sample with copper oxide growing between the amorphous surface layer on the left and the fcc substrate on the right. The ex-situ XPS data shown in Fig3a only demonstrate the growth of iron particles, not oxide, within the deformed matrix.

3. I am also concerned about the way the (S)TEM samples have been prepared. Focussed ion beam thinning leaves the surface of the samples in a very reactive state, so that even removing a sample of iron from the vacuum can lead to the nucleation of surface oxides before it can be transferred to the microscope. I appreciate that the authors say that they have minimised the transfer time but, in my experience, this may not be sufficient.

4. Could the authors explain why CuO and Fe₂O₃ are formed, in preference to other possible oxides of copper and iron?

5. The English used in the paper needs to be improved. In some places it even masks the message

that the authors are trying to convey. For example, p3 139. What does "no matter during powders consolidation" mean?

Reply to review comments

We are very grateful to all reviewers for giving constructive suggestions regarding our first version of manuscript. To answer the technical concerns, we have carried out many supplementary experiments, like XPS, EELS, Synchrotron X-ray Diffraction and execute DFT calculations. All concerns will be addressed in detail below. Some supplementary results are added to the new version of **Supplementary Information**. To avoid the possible confusion, all figures shown in this reply letter are indicated with additions of letter “R”, for example, Fig. R1.

Reply to Reviewer #1:

We really appreciate the reviewer for positive comments and helpful suggestions.

To address these comments and concerns, we have done comparative experiments to investigate the effects of altered layers on the TEM sample surfaces, and implemented systematical XPS studies regarding the oxygen contents in the initial and deformed materials. HRTEM image and EELS elemental mapping of *ex-situ* annealed sample were supplemented. All concerns will be addressed in detail as follows.

1. There is no information about how the TEM specimens were prepared. Was there an altered surface layer of the TEM specimens due to ion beam thinning? How thick was the damaged layer? How thick were the TEM specimens in the areas that were studied? Did the TEM specimen thickness correspond to the grain size of the materials?

Our TEM samples were prepared using mechanical thinning method combined with ion-milling. The detailed procedure is described as follows: The TEM samples were cut from the HPT disks, and all microstructural investigations were undertaken at radius of 3.0 mm from the torsional axis of the HPT deformed disks as shown in the following schematic diagram of Fig. R1 (to avoid the possible confusion, all figures shown in this reply file are indicated with additions of letter “R”, for example, Fig. R1). The TEM disk was then mechanically thinned and polished to a thickness of about 30 – 40 μm , followed by mechanical dimpling with a remaining thickness of about 15 μm in the center of the dimple. Subsequently the samples were ion-milled using a Gatan Precision Ion Polishing System 691 (Gatan, Inc., Pleasanton, USA) at -170 °C via liquid nitrogen cooling until perforation with voltage of 4 kV and angle of 4°, and then gentle polishing with lower limit conditions of 1.5 kV and 1° – 1.5° was implemented on the sample hole to

remove artefacts on the sample surfaces and make the thin area as flat as possible nearly without thickness gradient. The schematic diagram of ion-milling process is shown in Fig. R2.

Actually due to the sputtering of accelerated Ar ions on the sample surfaces during ion-milling, it seems to be inevitable to avoid the surface alteration. In addition, for TEM sample, the so-called surface relaxation phenomenon is existed at the thin area. Unfortunately, it is really hard to exactly know how thick of the surface altered layer, because it is related to the materials, internal strains and ion-milling parameters. However, as mentioned above, we have taken a measure by controlling the ion-milling parameters, with the lowest voltage of 1.5 kV and angle of $1^\circ - 1.5^\circ$ as long as a weak ion current can be detected in the equipment, to gently polish the sample surfaces at the final ion-milling stage. By this way, the effect from surface altered layer due to ion-milling can be minimized. For all TEM samples we prepared in this way, we didn't observe any altered surface layer in HRTEM images even at the extremely thin edges. If the influence from such layer is pronounced, the changed areas should be detected by HRTEM as damaged regions. Actually we have done comparative experiments using the same material as reported in the manuscript, HPT-deformed 75Cu-25Fe nanocrystalline alloy. We ion-milled the sample with parameters of voltage of 6 kV and angle of 8° for about 20 min until perforation, and then checked the sample in TEM. The BF and HRTEM images are shown in Fig. R3. It can be seen that the blurred spots with dimension of about 5 nm are damaged areas by ion hitting. Some spots are even changed to amorphous structures. However, a trial using ion-milling parameters of voltage of 5 kV and angle of 6° with liquid nitrogen cooling on as-deformed nanocrystalline pure Cu and 75Cu-25Fe showed no altered structures according to BF and HRTEM images, which are shown in Fig. R4. The grains shown in HRTEM images Fig. R4b and R4d are quite clear and clean without any blurred spots. These comparative experiments show that under ion-milling

conditions of 5 kV and 6°, no altered structures were introduced to the sample surfaces. Furthermore, the highest ion-milling parameters we used in this work are 4 kV and 4°, which should make the sample surfaces more stable and consistent with the internal structures.

The TEM sample thickness close to the edge is about 15 – 25 nm, and the average grain size of our investigated sample 75Cu-25Fe is about 56 nm which has already been mentioned in the manuscript. So the TEM sample thickness is much smaller than the average grain size, and it must be ensured that no grain overlaps exists in the area for *in-situ* HRTEM observation, otherwise Moiré fringes will be observed. But our HRTEM image recorded at 20 °C shown in the manuscript Fig. 2 is quite clear and doesn't display any Moiré fringe.

Fig. R1 Schematic diagram of HPT deformation and position of TEM sample derived from as-deformed HPT disk.

Fig. R2 Schematic diagrams of (a) top-view and (b) side-view of ion-milling process.

Fig. R3 (a, b) BF and (c, d) HRTEM images of as-deformed 75Cu-25Fe nanocrystalline alloy ion-milled with voltage of 6 kV and angle of 8° for about 20 min.

Fig. R4 (a, c) BF and (b, d) HRTEM images of as-deformed (a, b) pure Cu and (c, d) 75Cu-25Fe nanocrystalline alloys ion-milled with voltage of 5 kV and angle of 6° until perforation with liquid nitrogen cooling. The blue rectangular frames in BF images (a, c) marked the grains at the edge region which are shown in HRTEM images (b, d).

2. The authors claim that oxides are formed in the interior of the grains. However, the TEM images are projections of the TEM samples. The presented data do not exclude the possibility of the oxides forming at the TEM specimen surface. Further information is provided before the author can claim that the oxides are formed in the interior of the grains. This is crucial for the statements in the discussion and abstract.

We appreciate that this concern was raised by the reviewer. Actually from the HRTEM images shown in Fig. 2 in the manuscript, it can be determined that the oxides are formed inside the *fcc* matrix, rather than only on the surfaces. Probably the HRTEM images shown in the first version were too small to be distinguished clearly. Fig. R5 shows the HRTEM images extracted and enlarged from Fig. 2 in the first version of manuscript. We can see the image Fig. R5a recorded at room temperature is perfectly on zone axis of $[011]_{fcc}$ and quite clean without any blurred area. When the sample is heated to 180 °C, some areas get blurred, and indeed Moiré fringes can be seen. The mappings shown in Fig. 2 in the manuscript confirm that these blurred areas are CuO and Fe₂O₃. The occurrence of Moiré fringes means the overlaps of oxides and the *fcc* matrix, which is due to that the oxides are nucleating and developing at the low temperatures, and their sizes are quite small compared to the sample thickness of the observed area. As the temperature increased to 260 °C, it can be seen that the blurred areas expand and their structures get more ordered. Simultaneously, the Moiré fringes change to be less obvious compared to previous images recorded at 180 °C. For the image taken at 380 °C shown in Fig. R5d, Moiré fringes are barely seen and the oxide areas are even bigger, which imply that oxides have already developed to pass through the thickness direction of the sample and no overlaps exist anymore. Although the TEM images are 2D projections of the TEM samples, from the gradual disappearance of Moiré fringes, we can judge the development of oxides. Based on Fig. R5d, we can know the oxides fully formed inside the grains after annealing at high temperatures because of no observation of overlaps. Meanwhile, as shown in the Fig. 2 in the manuscript, we can roughly estimate that the dimension of oxides is about 10 – 20 nm, which is comparable to the sample thickness. Because we have proved in the first version of manuscript that the matching relationship between oxides and *fcc* matrix existed. If the oxides are only formed on the sample surfaces, definitely Moiré fringes will be observed (compared to those Cu oxides form on the

TEM sample surfaces shown in Fig. R11). From this point of view, we can judge the oxides are inside the materials. We also checked the *ex-situ* annealed sample by HRTEM as shown in Fig. R6, and for some area no Moiré fringe is observed, and morphology looks the same as already reported precipitates existed inside grains¹⁻⁴. According to the morphologies of oxides shown in HRTEM images of *in-situ* and *ex-situ* annealed samples, we can draw the conclusion that oxides are formed in the interior of grains. Synchrotron X-ray diffraction results shown in Fig. R12 will confirm this point.

An additional evidence to prove the oxide particles are growing within the sample, instead on the surface of the sample is to carry out the line-scan analysis crossing the oxide particle using EDXS/EELS (as shown in the following Fig.R8). The core intensity/signal at the oxide particles will give difference when the oxide particles form on the surface or grow within the sample.

Fig. R5 HRTEM images extracted and enlarged from Fig. 2 in the first version of manuscript, which were recorded at different temperatures during *in-situ* heating experiment, (a) 20 °C, (b) 180 °C, (c) 260 °C, (d) 380 °C.

Fig. R6 (a) HRTEM image of *ex-situ* annealed sample with inset of FFT image of the area marked with white frame. (b) HRTEM image extracted from the area marked with white frame in (a). (c) IFFT image showing oxide mapping corresponding to image (b) using FFT reflections marked with red spots in inset in image (a).

3. The morphology of the oxides are not shown in the ex situ annealed samples. Why?

The HRTEM image of oxide of *ex-situ* annealed sample can be referred to Fig. R6. Meanwhile the EELS mappings of *ex-situ* annealed sample will be shown below as requested in seventh question, where the morphologies of oxides and Fe precipitates can be clearly seen.

4. What was the partial oxygen pressure in the TEM during observation?

The column vacuum in our TEM machine JEOL 2100F is a high vacuum system with vacuum of about 5×10^{-7} Pa. If we regard the oxygen volume fraction is about 1/5 just like air, the partial oxygen pressure can be calculated to be 10^{-7} Pa.

5. Did the authors determine the oxygen content of the initial raw powders? Did it correlate to the oxygen content in the HPT materials?

We thank the reviewer for this suggestion. We have re-measured the oxygen contents in the initial raw powders, the as-deformed HPT sample and a piece of commercial pure Cu material (nominal purity of 99.99%, used as a reference) using X-ray photoelectron spectroscopy (XPS, ESCALAB 250Xi, Thermo Fisher Scientific, Waltham, USA). We have taken a series of special measures to remove the possible surface oxide layers before transfer the samples to the XPS chamber. All sample surfaces were fully polished in a media of ethyl alcohol and then transferred to XPS chamber immediately, which was followed by Ar ion sputtering with ion energy of 3 keV for 5 minutes to completely remove the possible surface oxide layers. We would emphasize that all samples were kept in ethyl alcohol after polishing, and the operation time of transfer from

ethyl alcohol to XPS chamber was controlled to a minimum of about a few seconds. All three samples mentioned above were transferred into the chamber at the same time.

Fig. R7a shows the XPS spectra in a full range of 0 – 1300 eV. Some typical peaks of Cu and Fe peaks are indexed on the profiles. To clearly check the O 1s peaks, the spectra within the region of 520 – 570 eV are enlarged as an inset in Fig. R7a, where O 1s peaks can be seen clearly for the 75Cu-25Fe as-deformed sample and the compacted powders while almost no signal can be observed for the commercial pure Cu sample at the position of 530 – 532 eV. The comparative study by XPS provides the direct evidence of oxygen existence in powder samples.

Fig. R7b shows the fine measurements of O 1s and Fe 2p_{3/2} peaks of 75Cu-25Fe as-deformed HPT sample and compacted raw powders respectively, and Fig. R7c shows the fine measurement of O 1s peak of commercial pure Cu rod. For the Fe 2p_{3/2} peak, it can be seen that every well resolved Fe 2p_{3/2} spectrum shows multiplet splitting with second component shifting to higher energy by 0.9 eV from the main peak. The position of main peak at 706.6 eV and the typical asymmetric peak shape containing a 0.9 eV higher component at 707.5 eV, both attest the Fe atoms are substantially in zero-valent states. If we compare the O 1s peak shown in Fig. R7c with the O 1s peaks of 75Cu-25Fe as-deformed HPT sample and compacted raw powders, we can see the signal-to-noise ratio of the O 1s peak of commercial Cu rod is very low and the peak almost can be negligible, which means probably only very tiny amount of oxygen exists in the pure Cu rod. Meanwhile, it proves that our measurements are effective without introducing adsorbed oxygen.

To evaluate the content of oxygen in each sample, peaks of O 1s, Cu 2p_{3/2} and Fe 2p_{3/2} were selected to quantify the integrated areas. Table R1 displays the quantification result of the contents of O, Cu and Fe in 75Cu-25Fe as-deformed sample, commercial pure Cu and 75Cu-

25Fe compacted powders, with considering relative sensitivity factors of 2.881 (O), 16.73 (Cu) and 10.7 (Fe). It can be seen the powder samples contain a level of about 3 at.% oxygen inside material while for commercial pure Cu the oxygen content is less than 1 at.%. It should be emphasized that for the fine scan spectrum of commercial pure Cu, the signal-to-noise ratio of O 1s peak is quite poor and it is hard to calculate the integrated area of this peak accurately, so the given value here of 1 at.% is guaranteed to be overestimated.

The measured oxygen content in 75Cu-25Fe as-deformed disk here is 3.6 at.%. Actually we have carried out another two independent measurements on two separate as-deformed samples, the oxygen contents in these two samples are 3.4 at.% and 3.3 at.% respectively. So it can be assured that the oxygen content in the HPT as-deformed samples is about 3.5 at.%. As for the reason that the measured oxygen value in the compacted powders is 3.0 at.%, a little bit lower than the value in samples after HPT deformation, it may be attributed to the pores formed during compaction which is segmented and the oxygen has reacted with metal elements during continuous deformation.

Synchrotron X-ray diffraction results shown in Fig. R12 and the result of *in-situ* heating experiment on the arc-melted bulk sample (as a reference to show the oxygen content difference) shown in Fig. R13 will testify the oxygen existence in the HPT deformed 75Cu-25Fe powder sample.

Fig. R7 (a) XPS measurements of 75Cu-25Fe as-deformed HPT sample (black spectrum), commercial pure Cu rod (red spectrum) and 75Cu-25Fe compacted raw powders (blue spectrum). (b) Fine measurements of O 1s and Fe 2p_{3/2} peaks of 75Cu-25Fe as-deformed HPT sample and compacted raw powders. (c) Fine measurement of O 1s peak of commercial pure Cu rod.

Table R1

Constituents of 75Cu-25Fe as-deformed sample, commercial pure Cu rod and 75Cu-25Fe compacted raw powders calculated based on XPS spectra.

Samples	O (at.%)	Cu (at.%)	Fe (at.%)
As-deformed HPT disk	3.6	74.9	21.5
Commercial pure Cu rod	< 1.0	–	> 99.0
Compacted raw powders	3.0	73.3	23.7

- Figure 3 in the supplemental material illustrate the presence of Fe grains at 420 °C. It is not clear how the authors reach the conclusion that the two grains are pure iron grains. Please clarify.

Fig. 3 shown in supplemental material of the first version displays the grain morphology change of 75Cu-25Fe sample during *in-situ* annealing by recording ADF images. After the sample was annealed at 420 °C, we carried out EDXS line scans on the newly-grown grains, so they can be determined to be Fe grains by EDXS line measurements just like the image and profile shown in Fig. 5 in supplemental material of the first version.

7. Figure 5 in the supplemental material shows concentration profiles after thermal anneal. The presented data do not verify that the formation of iron occurs as a result of the thermal anneal since there are no data provided before thermal anneal. A figure corresponding to Figure 3 c, in the main manuscript, with data from EELS mapping at 20 °C for both *in situ* and *ex situ* would provide crucial information for this manuscript.

As we mentioned above in the sixth question, EDXS line scan was implemented on the annealed sample across the grains to determine the newly-generated Fe grains. Actually, from the morphologies of the annealed sample, the Fe grains grown due to the decomposition usually have rounded corner shapes rather than the irregular multi-angular grains of the matrix, just as shown in the Fig. 3 and Fig. 5 in the supplemental material of the first version. Based on this experience, we first select such newly-grown grains with rounded corners and then confirm the Fe grains by EDXS measurements. The data of Fig. 3 shown in the supplemental material of the first version has displayed the occurrence of Fe grains due to the thermal annealing.

We thank the reviewer for suggesting us to carry out the EELS mapping for the *ex-situ* annealed sample for confirmation of the EELS mapping results of *in-situ* heating sample. Fig. R8 shows the EELS elemental mapping of *ex-situ* annealed sample at 420 °C. As we mentioned above, the Fe grains grown due to the decomposition usually have rounded corner shapes which can be also revealed from the Fe_L mapping in Fig. R8d. Actually the morphologies of Fe grains are almost the same as the *in-situ* annealing results, with grain size of about 20 – 50 nm. By comparing the Fe_L mapping, we can see the dark areas are corresponding to the Fe grains in Fig. R8a, where some small Fe grains are marked with white circles. Fig. R8b shows the O_K mapping, from which our conclusion of oxides forming during annealing with dimensions from

several nanometers to tens of nanometers is confirmed. Except some oxides precipitates existed at the grain boundaries, many smaller oxides formed inside the grains as observed by HRTEM images. The Fe concentration histogram shown in Fig. R8e was extracted along a white arrow in Fig. R8d from a pure Fe grain to the Cu matrix. It can be seen that after *ex-situ* annealing at 420 °C, the residual Fe in Cu matrix is about 18 – 20 at.% which is consistent with the result reported in Fig. 3c in the manuscript.

Fig. R8f is a line profile crossing Fe oxide formed within the sample. It is clearly shown that the Fe and O concentration reach maximum while Cu concentration is minimum. This is a strong indication that Fe oxide forms within the grain instead on the surface of the sample.

Fig. R8 EELS elemental mapping of *ex-situ* annealed sample at 420 °C. (a) HAADF-STEM image. (b) O_K mapping. (c) Cu_L mapping. (d) Fe_L mapping. (e) Fe concentration histogram obtained from line scan along red arrow displayed in (d). (f) A line profile crossing Fe oxide formed within the sample along the white arrow displayed in (b-d).

8. There are small areas with bright contrast at the grain boundaries in Figure 5 a in the supplementary material as well as the areas in the white circles. What is the difference between the small bright areas?

Fig. R9 shows the ADF-STEM image of 75Cu-25Fe after *ex-situ* annealing at 420 °C which is extracted from Fig. 5a in supplemental material of the first version. Some blue arrows indicate the grain boundaries which have different contrasts with neighboring grains. The contrast differences in grain boundaries are mainly caused by the slight tilt of grain boundaries which are not exactly “edge-on”, leading to grain overlapping. “Edge-on” means that the grain boundary is exactly parallel to the incident beam, which makes the grain boundary viewed as a fine line in the image rather than a widened blurred area as shown in the Fig. R9.

Fig. R9 ADF-STEM image of 75Cu-25Fe after *ex-situ* annealing at 420 °C which is extracted from Fig. 5a in supplemental material of the first version.

Reply to Reviewer #2:

We appreciate the reviewer's constructive comments very much.

One of the reviewer's main concerns is about the novelty of this work. To address this point and satisfy the referee, the main statements regarding the novelty are given as follows:

Nowadays, production of applicable bulk nanocrystalline alloys using severe plastic deformation is of great importance for the next-generation high-performance structural materials, and it is drawing growing interests in the materials research field and industry applications. Unfortunately one prominent problem arising during the consolidation and straining processes is the unavoidable contamination from gaseous species for powders-processing technology, such as oxygen. Currently, people have realized that oxygen contamination can induce large discrepancies in the mechanical properties, microstructures and thermal stabilities. However, the exact state of oxygen contaminant inside the materials remains unclear, and how oxygen atoms behave during annealing has hardly been explored. Moreover, developing new nanostructured bulk materials for potential applications demands atomic-resolution insights into the structures, particularly, the understanding of oxygen behaviors in nanostructured materials. In this work, we employed the modern spherical aberration corrected high-resolution transmission electron microscopy to observe the oxygen behavior via *in-situ* annealing at the atomic scale. The results revealed that except decomposition process, the nanometer-sized oxide clusters could form inside the grains at specific temperatures. i) This is a first atomic-scale observation of the oxidation and decomposition process in bulk nanocrystalline alloys; ii) The finding in this study is of great significance in developing a new route for designing new bulk nanocrystalline alloys by intentionally introducing different oxygen content, which can produce dispersive nano-sized oxides in the matrix so that the mechanical properties can be tuned as desired; iii) Actually, in

general, this study will assist understanding the oxygen atom behavior in other material systems, such as in nanostructured materials, oxides and steels, and clarifying its role in the field of surface science, and catalysts.

As for the other technical concerns, we have implemented systematic XPS studies regarding the oxygen impurity in the initial materials (powder material and arc-melted bulk materials), and recalculated the formation enthalpies using DFT + U model with consideration of the strong correlation effects between transition metal elements and oxygen. In addition, we quantified the electron density under the HRTEM imaging condition. The concerns will be addressed in detail below.

1. The authors have missed out on a large part of the rather rich literature of oxidation and oxidation kinetics in oxide dispersion strengthened (ODS) steels. These are mentioned very briefly in the introduction but no reference is made to significant studies of kinetics, phase formation, effect of ion beams (or electron beams) on the evolution of these nanostructured alloys.

We thank the reviewer for helpful suggestion. Some typical literature related to ODS materials are included in the revised version of the manuscript, and some descriptions related to ODS alloys are modified. The reason that we mentioned the ODS steels in the introduction part is that we tend to extend the idea on our observed intragranular formed oxides in nanocrystalline materials for the other potential application, similar like the ODS steels.

2. The authors give too little information on the impurity levels and characteristics in the as-received materials. It is not trivial to ascertain if low-T oxidation is abnormal or curious, as the authors claim, when we have no clear information on the impurity contents, other

than the balance numbers. This information should be available to the authors and will be very useful when drawing conclusions.

We appreciate the reviewer's comments very much on determining the impurity level in the raw materials. We have done systematic investigations on the impurity contents in raw powders and as-deformed HPT samples using XPS. As we replied to the fifth question (question # 5) raised by the first reviewer about the impurity, in the light of prudent measurements using XPS, we can determine that in our HPT deformed Cu-Fe samples from powders, the oxygen impurity level is 3.5 at%. We know the nanocrystalline alloys deformed by high pressure torsion by extremely large strains are in high non-equilibrium status. A large number of defects and boundaries are existed inside the materials, resulting in very high interfacial energy, elastic energy and so on, which is the reason for the thermally instability of the high-strained nanostructured alloys. As we discussed in the manuscript, the high stored energy in non-equilibrium conditions should be responsible for the low-temperature oxidation and decomposition behaviors in the severely deformed materials.

Synchrotron X-ray diffraction results shown in Fig. R12 and the result of in-situ heating experiment on the arc-melted bulk sample (as a reference to show the oxygen content difference) displayed in Fig. R13 will testify the oxygen existence in the HPT deformed 75Cu-25Fe powder sample.

3. The language of the supplementary materials has to be improved significantly. There is a large discrepancy between the manuscript and the suppl.mater. There are also some small

issues with the manuscript itself, but these are minor and should be caught by a third party that proofs it.

We are grateful to reviewer's suggestions. The language in the revised version of manuscript has been checked by the native-English speaker, so it has been largely improved.

4. The very weak O 1s signal in the suppl.mater. fig 2 providing an estimate on the composition is far from solid. No error bar or uncertainty is presented either, which is here rather glaring.

As shown in previous replies regarding to the oxygen contents in the materials, we have carried out a series of careful investigations on the raw materials, as-deformed samples as well as commercial high purity pure Cu rod to quantify the oxygen contents inside the materials using XPS. To exclude the possibility of surface oxide layers on the sample surfaces, some special measures have been taken to remove the surface layers during polishing and sample transfer, which have been described in detail in the fifth question (Question #5) raised by the first reviewer. It should be noted that all the mentioned samples were transferred into the XPS chamber at the same time and kept at the same condition. In addition, we have implemented three independent measurements on separate samples using the same procedures mentioned above, and the oxygen contents are 3.4 at.%, 3.3 at.% and 3.6 at% respectively. Therefore, these data can sufficiently support the conclusion of that oxygen content in the as-deformed sample is (3.43 ± 0.15) at% indeed. For simplification, it is written as 3.5 at.% in the as-deformed materials.

As for the concern of that such relatively high oxygen content of 3.5 at.% in Cu-Fe solid is far from the equilibrium solution amount, the explanation is that our materials were generated under

the high pressure torsion and Cu-Fe systems formed the single phase supersaturated solid solutions which were far from the conventional equilibrium states. Meanwhile, the oxygen atoms were dissolved into the *fcc* matrix, being assumed to occupy the octahedral interstices, which were also in high non-equilibrium states. Therefore, here, the oxygen existing in deformed alloys is different from the oxygen present in the conventional equilibrium solid solutions.

5. Vanilla DFT applied on transition metal oxides is a well-known problem area. Strong correlation effects can be quite important. The use of the enthalpies here thus calculated is not so significant in the current manuscript but still, one should do this properly or not at all.

In the first version of manuscript, the reason why we use DFT is to calculate the formation energies of likely oxides. We attempt to explain the formation of CuO and Fe₂O₃ rather than other kind of oxides from the viewpoint of formation energies.

Due to the strong correlation effects in transition metal oxides, PBE⁵ exchange-correlation function with LDA and GGA pseudo-potentials and Hubbard U model^{6,7} were considered. Here, for Cu and Fe, U = 3 eV was employed. The results are shown in Table R2, which are in reasonable agreement with literature⁸. Although the strong correlation of transition metal oxides, combining our calculated results with the values given in literature⁸, we can draw the conclusion that CuO and Fe₂O₃ possess the lowest formation energies in Cu and Fe oxides.

Table R2

Calculated formation enthalpies for possible different oxides (eV/atom).

	LDA	LDA+U	GGA	GGA+U
Cu ₂ O	-0.703	-0.767	-0.599	-0.666
CuO	-1.091	-1.086	-0.873	-0.874
Fe ₂ O ₃	-1.947	-1.980	-1.523	-1.563
FeO	-1.257	-1.745	-1.171	-1.576

6. The study in the suppl.mater motivating the claim in the main manuscript that the e-beam has no effect is far from conclusive. I would recommend the authors to improve the quantitative analysis of this effect (or lack thereof).

We appreciate very much that the reviewer commented on beam effect. As we mentioned in the manuscript, actually it is well known that in TEM studies the electron beam effect is non-negligible, and should be considered because it may generate extra heat and facilitate the chemical reaction process. Especially in *in-situ* experiments, electron beam effects are unavoidable and should be taken into consideration.

During our experiments, we have taken some measures to minimize the potential influences of the electron beam. I) During image recording, the electron beam was spread into a specific size fitting completely to the fluorescent screen every time, and the beam was switched off during the heating process and the imaging was done immediately within 10 s after heating was finished. As pointed out by the reviewer, we have quantified the electron density on two images with magnifications of 600K and 800K respectively, recorded on the same area as shown in Fig. R10. The two insets are distributions of the number of pixels versus values of counts. The green coverage areas shown in two HRTEM images correspond to the majorities of pixel intensities

within a region shown in two insets. With the exposure time of 1.0 s and beam spreading to fit to the size of fluorescent screen, the electron density of image taken at 600K is $6405 / \text{\AA}^2 \cdot \text{s}$ while for image with the magnification of 800K this value is $11172 / \text{\AA}^2 \cdot \text{s}$. For our JEOL2100F equipment at 200kV, 4 counts can be converted to be equivalent to 1 electron. Generally, the larger magnification will get higher electron density because of the more convergent beam used at the higher magnification. At the imaging conditions as above-mentioned, the current density is about 68.8 pA/cm^2 for 600K and 69.2 pA/cm^2 for 800K, respectively.

Actually, in some *in-situ* TEM investigations about the chemical reactions, people have already realized the beam effects. For example, the diffusion of oxygen in Bi_2O_3 can be enhanced due to the beam effects⁹. It is also mentioned although beam effects are unavoidable in the electrochemical cell TEM experiments since an electron beam is a necessary source for imaging, under specific imaging conditions with moderate electron current density no obvious precipitation or cluster is observed¹⁰. On the other hand, due to the thin TEM sample, with thickness of only about tens of nanometers, heat dissipation proceeds very fast which will not generate heat accumulation. The results shown in Fig. 4 in supplementary material (comparative experiments) confirm that in current investigated systems the oxidation and decomposition/precipitation processes are not relevant to the beam effect. That is to say, beam effects can be negligible in this work.

Fig. R10 Electron density analyses of HRTEM images of the same area taken at (a) 600K and (b) 800K. The insets are distributions of the number of pixels versus values of counts.

Reply to Reviewer #3:

We appreciate the reviewer's constructive comments very much.

One of the reviewer's main concerns is about the novelty of this work. In the revised version, we made this clear by rewriting the Abstract and Introduction parts to sort out the main information, and emphasize the importance of his study. To address this point and satisfy the referee, the main statements regarding the novelty are given as follows:

Nowadays, production of applicable bulk nanocrystalline alloys using severe plastic deformation is of great importance for the next-generation high-performance structural materials, and it is drawing growing interests in the materials research field and industry applications. Unfortunately one prominent problem arising during the consolidation and straining processes is the unavoidable contamination from gaseous species for powders-processing technology, such as oxygen. Currently, people have realized that oxygen contamination can induce large discrepancies in the mechanical properties, microstructures and thermal stabilities. However, the exact state of oxygen contaminant inside the materials remains unclear, and how oxygen atoms behave during annealing has hardly been explored. Moreover, developing new nanostructured bulk materials for potential applications demands atomic-resolution insights into the structures, particularly, the understanding of oxygen behaviors in nanostructured materials. In this work, we employed the modern spherical aberration corrected high-resolution transmission electron microscopy to observe the oxygen behavior via *in-situ* annealing at the atomic scale. The results revealed that except decomposition process, the nanometer-sized oxide clusters could form inside the grains at specific temperatures. i) This is a first atomic-scale observation of the oxidation and decomposition process in bulk nanocrystalline alloys; ii) The finding in this study is of great significance in developing a new route for designing new bulk nanocrystalline alloys by

intentionally introducing different oxygen content, which can produce dispersive nano-sized oxides in the matrix so that the mechanical properties can be tuned as desired; iii) Actually, in general, this study will assist understanding the oxygen atom behavior in other material systems, such as in nanostructured materials, oxides and steels, and clarifying its role in the field of surface science, and catalysts.

To address all the technical concerns and consolidate *in-situ* HRTEM observations, we have performed systematic XPS studies regarding the oxygen contents in the initial and deformed materials, and synchrotron X-ray diffraction measurements on the powder sample and reference sample (arc-melted bulk material, supposed to be less oxygen content) with the same Cu-Fe composition under the identical measurement conditions. Most importantly, the *in-situ* heating experiment was carried out also on the reference sample (arc-melted bulk sample) with recording HRTEM images, being used as a comparative atomic-resolution experiment to address the effect difference when the different oxygen contents are involved in the two materials (powder sample and reference sample).

All other concerns will be addressed in detail as follows.

1. The oxygen analysis by XPS is not very convincing. I would be much more convinced if a piece of pure copper had been ion cleaned and analysed at the same time. Why was a quantitative bulk analysis technique not used to determine the oxygen content of the samples?

We thank the reviewer to point out the comparative method of the XPS measurement. As we replied for the fifth concern raised by the first reviewer, we have done a series of investigations to

systematically study the oxygen content in the as-deformed Cu-Fe alloys. The re-measured spectra and calculated constituents are shown in Fig. R7 and Table R1. Based on the different measurement results, it can be determined that the oxygen content in the HPT deformed Cu-Fe alloy is about 3.5 at.%.

As we mentioned before, for the XPS measurements, the most important point is to exclude the influence from the possible surface oxide layers. We have taken a series of measures to keep the sample surfaces unoxidized. We polished all the samples in the ethyl alcohol and then transferred to XPS chamber immediately at the same time. The sample surfaces were then severely sputtered by Ar ions with energy of 3 keV for 5 minutes to completely remove the possible surface oxide layers. By such measures, we can check all the samples at the same conditions and the results are accurate by our comparative study. It is very ideal to measure the oxygen content using quantitative bulk analysis techniques. Unfortunately, we could not find a proper bulk analysis technique which can fully exclude the possible influence of the surface oxide layers. Another advantage for XPS measurement is that we can obtain not only the constituents but also the valence states of the elements, by which their chemical environments corresponding to deformation behaviors can be judged.

2. What is the evidence that the oxide particles are growing within the sample during the in-situ heating and not on the surface of the sample. Indeed the micrograph shown in Fig. 4a looks like a section through the edge of a thin sample with copper oxide growing between the amorphous surface layer on the left and the fcc substrate on the right. The ex-situ XRD data shown in Fig3a only demonstrate the growth of iron particles, not oxide, within the deformed matrix.

The same concern was raised by the first reviewer in the second question (question #2), and we have addressed it clearly from the point of view of formation of Moiré fringes. In addition, we have done a comparative study to show that if any oxides form only on the sample surfaces, Moiré fringes will be definitely observed. Fig. R11 shows HRTEM and FFT images of pure Cu foils exposed to air for a certain time and TEM sample of as-deformed 75Cu-25Fe alloy exposed to air for 2 days. From the FFT images, we can know that CuO layers formed due to the exposure to the air. The Moiré fringes indicate that the oxide layers have adhered to the sample surfaces. The morphologies of oxides are absolutely different from those oxides grown within the sample during *in-situ* heating as shown in Fig. R5. An additional evidence to prove the oxide particles are growing within the sample, instead on the surface of the sample is to carry out the line-scan analysis crossing the oxide particle using EDXS/EELS (for instance, Fig.R8). The core intensity/signal at the oxide particles will give difference when the oxide particles form on the surface or grow within the sample.

Fig. 4a in the first version of manuscript actually was taken from the *in-situ* annealed sample at the highest temperature of 420 °C. As we mentioned in the manuscript, oxygen will move inside the matrix lattice during heating, and it is possible for the oxygen to gather at the edge area via diffusion where the surface energy is high. From the morphology of the CuO shown in HRTEM image, it is more likely that a small edge part of the [011] grain underwent a chemical reaction from Cu to CuO during heating. Because no Fe and Fe₂O₃ precipitates were mixed together at the thin area, we got opportunity to observe the atomic structures and the matching relationship between CuO and *fcc* matrix.

As for the XRD profiles shown in the Fig. 3a for the *ex-situ* annealed samples in the first version of manuscript, it was mainly for displaying the process of Fe decomposition. Actually,

we have tried to use the normal X-ray diffraction to detect the oxide formation process. But, in fact it is impossible because the signal-to-noise ratio was not high enough to indicate the trace amount of oxides present inside the material. Instead, we measured the *ex-situ* annealed samples using synchrotron method which has an extremely powerful energy to generate peaks even for the tiny amount components (as shown below). Synchrotron experiments were performed at the PETRA III P07 beamline at the DESY Photon Science facility (Hamburg, Germany).

First, we fabricated a reference sample by arc-melting with the composition of 75Cu-25Fe using the high purity commercial Cu and Fe rods (nominal purity: 99.99%). Because large pieces of Cu and Fe rods were used in arc-melting process, it could effectively reduce the influences of the surface oxides. Second, the powder sample and arc-melted sample with composition of 75Cu-25Fe were deformed by HPT to the same strains, followed by *ex-situ* annealing in Ar atmosphere at 420 °C at the same time. Then these two samples were cut and polished to the same shape and thickness. Synchrotron measurements were implemented on these two samples at the same conditions. Fig. R12a shows the synchrotron profiles of *ex-situ* annealed 75Cu-25Fe samples at 420 °C, which were deformed from powders and arc-melted bulk, respectively. Fig. R12b shows the images of powder and arc-melted bulk samples with blue circles indicating the measurement areas. From the synchrotron profiles, we can see that except all Cu and Fe peaks locating at almost the same positions, for the profile of the powder sample, some extra peaks appear at the left side of $(111)_{\text{Cu}}$ peak as well as between $(200)_{\text{Cu}}$ peak and $(200)_{\text{Fe}}$, which fit with peaks of CuO and Fe_2O_3 very well. The obvious peak with spacing of 2.529 Å is indexed as $(002)_{\text{CuO}}$, and the peaks at positions of 1.603 Å and 1.475 Å can be indexed as $(120)_{\text{CuO}}/(122)_{\text{Fe}_2\text{O}_3}$ and $(124)_{\text{Fe}_2\text{O}_3}$ respectively. We used the arc-melted sample with the same composition as a reference, and the oxides can be only detected for the powder sample. So the synchrotron measurements

provide a strong evidence to prove that the oxides formed inside the sample after annealing. The detailed results and discussion of the difference between powder sample and arc-melted bulk sample will be shown in our next paper.

Fig. R11 HRTEM and FFT images of (a) pure Cu foils exposed to air for more than 3 months and (b, c) TEM sample of as-deformed 75Cu-25Fe alloy exposed to air for 2 days.

Fig. R12 (a) Synchrotron profiles of *ex-situ* annealed 75Cu-25Fe samples at 420 °C, which were deformed from powders and arc-melted bulk respectively. (b) Images of powder and arc-melted bulk samples with blue circles indicating the measurement areas.

- I am also concerned about the way the (S)TEM samples have been prepared. Focussed ion beam thinning leaves the surface of the samples in a very reactive state, so that even

removing a sample of iron from the vacuum can lead to the nucleation of surface oxides before it can be transferred to the microscope. I appreciate that the authors say that they have minimised the transfer time but, in my experience, this may not be sufficient.

Thanks for the concern raised by the reviewer. The same question was also asked by the first reviewer in the first question (Question #1). Our TEM samples were prepared using mechanical thinning method, and combined with low voltage ion-milling as a final step. As we mentioned in the reply to the first reviewer, we have done a series of experiments to minimize the influence from the ion-milling process. After the final gentle polishing by ion-milling with the parameters of lower limit conditions of 1.5 kV and $1^\circ - 1.5^\circ$, we didn't see any clear ion damage from the HRTEM image, as shown in Fig. R4 (c, d).

As for the concern of surface nucleation of oxides or adsorption of oxygen, we agree the reviewer's opinion of that oxygen may somewhat adsorb on the surfaces of the sample as long as the sample is exposed to the air once, and it seems that it is inevitable during the entire experiments. However, what we are certain is that the influence from the surface oxygen adsorption can be very tiny on the *in-situ* heating experiment, and it doesn't affect the conclusion drawn from the current experiment. Our sample may be exposed to air only for 3 minutes (sometimes, less time 3 mins) during the transfer. Meanwhile, we did a test experiment to leave the sample inside the microscope for more than 12 hours, and then checked the sample with HRTEM, we didn't see any nucleated oxides or Moiré fringes as shown in Fig. R11. In addition, to confirm our conclusion, we carried out another comparative experiment like synchrotron measurements using above-mentioned 75Cu-25Fe arc-melted bulk material. We did the *in-situ* heating experiment on the 75Cu-25Fe alloy deformed from arc-melted bulk materials (which has less oxygen contents). Here we should emphasize that the TEM sample preparation method is

strictly the same as previous powder sample, and transfer time is controlled to be almost the same as previous experiment with about 3 minutes. Fig. R13 shows the HRTEM and corresponding FFT images of *in-situ* heating experiment on the 75Cu-25Fe sample deformed from arc-melted bulk. The selected grain is on [011] zone axis. We can see that all FFT images of the corresponding HRTEM images recorded at different temperatures, are quite clean and only main spots belonging to [011] zone axis can be observed. If we compare the images (shown in Fig. R13) to the images from *in-situ* heating of powder sample displayed in Fig. 2 in the first version of manuscript, the differences are obvious. For the FFT images of powder sample, there are several tiny spots of oxides appear even when the temperature reaches 100 °C. Therefore, the comparative *in-situ* heating experiment of arc-melted bulk sample is another evidence to show the influence of oxygen. This result can help to almost exclude the influence of adsorbed oxygen on the TEM sample surfaces.

Fig. R13 HRTEM and corresponding FFT images of *in-situ* heating experiment on the 75Cu-25Fe sample deformed from arc-melted bulk.

4. Could the authors explain why CuO and Fe₂O₃ are formed, in preference to other possible oxides of copper and iron?

For this concern, we have explained in the manuscript in terms of the formation energy. We have carried out DFT calculations. The theoretical calculations show that in all possible Cu and Fe oxides, CuO and Fe₂O₃ possess the lowest formation energy, which means CuO and Fe₂O₃ are easy to form as compared to other oxides. Our calculations give approximately the same values for the formation energy of oxides as previously reported⁸. For the first version of the manuscript, the second reviewer pointed out that our calculations should consider the strong correlation effects of transition metal oxide. So, we improved our models with considering PBE exchange-correlation function with LDA and GGA pseudo-potentials and Hubbard U model. The recalculated formation energies of oxides are listed in Table R2.

From another point, the synchrotron measurement also confirmed the formation of CuO and Fe₂O₃ after annealing. The CuO and Fe₂O₃ possess monoclinic and hexagonal structures respectively, while Cu₂O and FeO have cubic structures, and the lattice parameters of these oxides vary a lot. So it is easy to distinguish these oxides from plane spacing on both FFT, diffraction patterns and synchrotron spectra.

5. The English used in the paper needs to be improved. In some places it even masks the message that the authors are trying to convey. For example, p3 l39. What does “no matter during powders consolidation” mean?

We really appreciate the reviewer’s suggestion. The language of the resubmitted version of manuscript has been checked by the native-English speaker. We hope that our meanings can be conveyed clearly in the revised version.

References:

1. Jiang, S. *et al.* Ultrastrong steel via minimal lattice misfit and high-density nanoprecipitation. *Nature* **544**, 460–464 (2017).
2. Hirata, A. *et al.* Atomic structure of nanoclusters in oxide-dispersion-strengthened steels. *Nat. Mater.* **10**, 922–926 (2011).
3. Ribis, J. & De Carlan, Y. Interfacial strained structure and orientation relationships of the nanosized oxide particles deduced from elasticity-driven morphology in oxide dispersion strengthened materials. *Acta Mater.* **60**, 238–252 (2012).
4. Chen, J. H. *et al.* Atomic pillar-based nanoprecipitates strengthen AlMgSi alloys. *Science* **312**, 416–419 (2006).
5. Perdew, J. P., Burke, K. & Ernzerhof, M. Generalized gradient approximation made simple. *Phys. Rev. Lett.* **77**, 3865–3868 (1996).
6. Anisimov, V. I., Zaanen, J. & Andersen, O. K. Band theory and Mott insulators: Hubbard U instead of Stoner I. *Phys. Rev. B* **44**, 943–954 (1991).
7. Cococcioni, M. & De Gironcoli, S. Linear response approach to the calculation of the effective interaction parameters in the LDA+U method. *Phys. Rev. B* **71**, 035105 (2005).
8. Stevanović, V., Lany, S., Zhang, X. & Zunger, A. Correcting density functional theory for accurate predictions of compound enthalpies of formation: Fitted elemental-phase reference energies. *Phys. Rev. B* **85**, 115104, (2012).
9. Niu, K. Y., Park, J., Zheng, H. & Alivisatos, A. P. Revealing bismuth oxide hollow nanoparticle formation by the Kirkendall effect. *Nano Lett.* **13**, 5715–5719 (2013).
10. Zeng, Z. *et al.* Visualization of electrode-electrolyte interfaces in LiPF₆/EC/DEC electrolyte for lithium ion batteries via in situ TEM. *Nano Lett.* **14**, 1745–1750 (2014).

Reviewers' Comments:

Reviewer #1:

Remarks to the Author:

It is clear from the document entitled "Reply to review comments" that the authors have made a significant effort to answer our questions and to clarify the text and figures. This much appreciated. There is a focus in the manuscript on the claimed observation of intragranular oxide formation as evident from the abstract. My comments below therefore concentrate on the discussion concerning the intragranular oxide formation. Based on the open issues, I cannot recommend a publication in Nature Communications at this stage.

The authors claim that HRTEM figure in Fig. R5 (a) shows a region that is perfectly on zone axis and quite clean without any blurred area (page 7 in "Reply to review comments"). The figure shows contrast variations and also differences in fringe visibility in different regions depending on their orientation. This indicates that the entire area is not perfectly on zone axis.

The presence of Moiré fringes does not provide information about whether the oxide is formed on the surface of the TEM sample or within the TEM sample. It should also be noted that Moiré fringes are not always visible even if two or more crystalline regions/structures are overlapping along the path of the electron through the TEM specimen.

On page 7, in "Reply to review comments", line 7 from the bottom of the page, the authors say "Based on Fig. R5d, we can know the oxides fully formed inside the grains after annealing at high temperatures because of no observations of overlaps.". However, Fig. R5 (d) shows Moiré fringes indicating that there is overlap.

The ADF-STEM image in Fig. R9 shows bright contrasts at grain boundaries that seem to be oriented parallel to the incident electron beam direction (for example, upper right area). There are other examples where the grain boundaries have brighter contrast similar to the bright areas in the larger white circle. It cannot be excluded that there are oxide areas at the grain boundaries. In addition, there are areas of intergranular regions visible in Fig. R3 (d), right side upper and lower. These are located in the interior of the TEM sample and there are crystalline areas with lattice fringes, with different orientations, visible on both sides of the intergranular region. I am not convinced that the oxides are nucleated in the interior of the grains of these alloys based on the results presented here. The oxides can have been nucleated on the TEM samples surfaces and at the grain boundaries.

Reviewer #2:

Remarks to the Author:

The authors have made a good effort at answering the questions I and the other referees raised, but I still do not agree on this being novel or innovative enough to merit publication in Nature Communications. The justification given by the authors as response to this point are not sufficiently convincing. There is no doubt this merits publication, just not in this journal, in my honest opinion. I leave it to the editors to decide the matter.

Side note: In case of publication here or elsewhere, the re-written introduction would need some touching up language-wise. Phrases like "For decades, people usually think..." are not really up to standard for scientific literature.

Reviewer #3:

Remarks to the Author:

The authors have made an effort to clarify most of the points raised by the referees. Hence, in my opinion, the paper could now be published. However, I have two further observations, which should be considered:

1. The presence of moiré fringes does not prove that the oxides are growing on the surface of the sample. Any two overlapping lattices could give rise to sets of moiré fringes. Clearly if most of the through thickness of the sample is occupied by one oxide particle, then fringes will not be observed, but internal particles surrounded by matrix could still give rise to moiré fringes.

N.B. Note the lower case 'm' in moiré – it is not a Proper Noun.

2. Although the new references to ODS steels are a useful addition to the paper, the authors omit some of the most recent and relevant papers on the combined effects of deformation and recrystallisation on the dissolution and re-precipitation of oxide particles.

See, for example, some of the work of the Liverpool group:

K. Dawson, S. Haigh, G. J. Tatlock and A. R. Jones, Nano-Particle Precipitation in Mechanically Alloyed and Annealed Precursor Powders of Legacy PM2000 ODS Alloy, *J Nuclear Materials* 464, 200-209 (2015);

K. Dawson and G. J. Tatlock, The Influence of Deformation, Annealing and Recrystallisation on Oxide Nanofeatures in Oxide Dispersion Strengthened Steel, *J Nuclear Materials* 486, 361-368 (2017).

Reply to Review Comments:

We are very grateful to all reviewers for giving further constructive comments on our revised version of manuscript (NCOMMS-17-04523A). To well answer the technical concerns raised by referees, we provided new experiments data via atom probe tomography experiments and further revise the manuscript accordingly.

Reply to Reviewer #1:

We really appreciate the reviewer for pointing out the open issues. To address the main query on “oxides forming inside the grains instead on the surface”, we have conducted atom probe tomography (APT) and STEM experiments, which can strengthen our novel observations and clarify these technical concerns very clearly. As the APT experiments were quite time-consuming to be carried out, we were not able to include in the previous revised version within the limited time. In addition, to present the work more accurate and consistent, we slightly modified some descriptions in the text.

1. The authors claim that HRTEM figure in Fig. R5 (a) shows a region that is perfectly on zone axis and quite clean without any blurred area (page 7 in “Reply to review comments”). The figure shows contrast variations and also differences in fringe visibility in different regions depending on their orientation. This indicates that the entire area is not perfectly on zone axis.

The word “perfectly” is not rigorous at all. We accept reviewer’s criticisms. Some areas in the grain show contrast variations, due to the slightly zone-axis off (orientation changes). Several reasons could cause contrast variations and differences in fringe visibility in different regions: I) Ar ion-milling could cause inhomogeneously sputtering locally.

Therefore, thickness change causes the slight contrast variations. II) Local distortion (originated from the defects, e.g. dislocations, point defects, stacking faults etc...) leads to contrast variations and local orientation change. In addition, heating enhance the defects mobility, which can significantly affect the local contrast variations and fringe visibility in different regions within one grain.

During the *in-situ* heating experiments, we usually tracked several grains, and selected the orientation-unchanged grains to record HRTEM images for analyzing the oxidation and decomposition behavior. Once RTEM images were taken over a large region, it may still contain some small areas slightly off the zone-axis, showing contrast variations and differences in fringe visibility.

2. The presence of Moiré fringes does not provide information about whether the oxide is formed on the surface of the TEM sample or within the TEM sample. It should also be noted that Moiré fringes are not always visible even if two or more crystalline regions/structures are overlapping along the path of the electron through the TEM specimen.

Thank the reviewer for very helpful comments.

It is true that the presence of Moiré fringes hardly provide information on whether the oxide is formed on the surface or within the TEM sample. Here, with the new APT data available, two approaches were proposed to identify the oxides location.

- 1) We supplemented atom probe tomography (APT) experimental data to show the oxygen distributions after *ex-situ* annealing at 300 °C. The APT results are included in the revised version of manuscript (Fig. 3d, also shown below). From the upper overall image, some Fe-rich areas (green color) embedded with oxygen atoms are

observed with dimensions of 20 – 50 nm which accords with the results obtained from EELS mapping displayed in Fig. 1c. The bottom image shows that oxygen atoms distribute almost homogeneously except that some O-rich clusters with sizes of 3 – 8 nm form within the TEM sample after *ex-situ* annealing. The APT provides a direct evidence to display the oxygen dissolution and oxygen clusters.

Fig.3d. APT overview of the *ex-situ* annealed 75Cu-25Fe sample at 300°C and oxygen map highlighting O-rich clusters.

2) The second evidence is from the STEM-EELS mappings which can help to indirectly attest the oxides forming inside TEM sample. One line-profile is made crossing Fe-O particles (as indicated by white line), supplementary Fig. 6f (also shown below), Fe, O and Cu atom distributions are clearly shown. Note that the Cu concentration across the Fe-O particle decreases down to less than 10 at.%, approaching to ~ 0 when at the middle of particle location. If an oxide particle with a certain thickness is formed on the surface of TEM sample, the Cu concentration on the particle will not reduce to be ~ 0 at the middle location simple because of Cu substrate contribution. The line profiles in supplementary Fig.6 also confirm that the oxides formed within TEM sample. In addition, one may recognize that some big particles are formed at the grain boundary , and some small particles are nucleated in the interior of grains

Figure 6 (supplementary), EELS elemental mapping of *ex-situ* annealed sample at 420 °C. (a) HAADF-STEM image. (b) O_K mapping. (c) Cu_L mapping. (d) Fe_L mapping. (e) Fe concentration histogram obtained from line scan along red arrow displayed in (d). (f) Cu, Fe and O composition profiles extracted from line scan of the same position indicated by the white arrows in (b-d).

3. On page 7, in “Reply to review comments”, line 7 from the bottom of the page, the authors say “Based on Fig. R5d, we can know the oxides fully formed inside the grains after annealing at high temperatures because of no observations of overlaps”. However, Fig. R5 (d) shows Moiré fringes indicating that there is overlap.

Thank the reviewer for pointing out the issues.

As mentioned earlier, the presence of Moiré fringes hardly provide information on whether the oxide is formed on the surface or within the TEM sample, as well as inside the grains or at the grain boundary. Therefore, we used the newly-acquired APT data and STEM-EELS maps to justify the location of oxides. We modified the text accordingly.

4. The ADF-STEM image in Fig. R9 shows bright contrasts at grain boundaries that seem to be oriented parallel to the incident electron beam direction (for example, upper right area). There are other examples where the grain boundaries have brighter contrast similar to the bright areas in the larger white circle. It cannot be excluded that there are oxide areas at the grain boundaries. In addition, there are areas of intergranular regions visible in Fig. R3 (d), right side upper and lower. These are located in the interior of the TEM sample and there are crystalline areas with lattice fringes, with different orientations, visible on both sides of

the intergranular region. I am not convinced that the oxides are nucleated in the interior of the grains of these alloys based on the results presented here. The oxides can have been nucleated on the TEM samples surfaces and at the grain boundaries.

We appreciate the reviewer for raising the questions.

I) For nanocrystalline alloys, the small bright dots inside grains with different contrasts in STEM images could be due to many reasons, such as defects (dislocations and stacking faults), precipitates, ion-milling damages, and surface oxides etc... For Cu-Fe, here, these bright dot contrasts are due to the numerous dislocations and stacking faults.

II) As for the prior Figure R9 mentioned by the reviewer, actually it is difficult to judge which reasons cause the bright contrasts at grain boundaries only based on one ADF-STEM image. To address this query, we tracked this TEM sample and cleaned the sample completely by re-ion-milling for 1.0 h. After ion-milling the sample was transferred to the microscope within 3 minutes to minimize the possible surface oxidation. We recorded a series of BF, ADF, HAADF images (as shown below) at several thin areas (grain overlapping less likely). From these images, contrast variations at the grain boundaries were not observed. The following BF, ADF, HAADF images were recorded from two different positions after heavily cleaning of TEM sample (from the *ex-situ* 420 °C-annealed specimen. Via EDXS measurements, the dark areas marked by white arrows in HAADF images can be identified as Fe-rich particles.

Although these, it cannot be excluded that there are oxide areas at the grain boundaries. Fortunately, with the APT data and STEM-EELS maps available, as shown in Question 2, it becomes clear. In short, we concluded that some of oxides are nucleated in the interior of grains, and some of them are formed at the grain boundary.

III) As for the Figure R3d in prior 'reply to review comments', it is taken from the as-deformed 75Cu-25Fe sample without any heat treatment, and the TEM sample was intentionally subject to heavily ion-milling with a voltage of 6 kV and an angle of 8° for 20 min. So, there are a lot of damaged areas introduced, like amorphous intergranular regions embedded with 5 nm-sized crystallites, also including partially surface oxides. Because this sample was solely used for demonstrating the influence of ion-milling process, no special measures were taken to avoid the surface oxidation.

Reply to Reviewer #2:

The authors have made a good effort at answering the questions I and the other referees raised, but I still do not agree on this being novel or innovative enough to merit publication in Nature Communications.

The justification given by the authors as response to this point are not sufficiently convincing. There is no doubt this merits publication, just not in this journal, in my honest opinion. I leave it to the editors to decide the matter.

Side note: In case of publication here or elsewhere, the re-written introduction would need some touching up language-wise. Phrases like "For decades, people usually think..." are not really up to standard for scientific literature.

We are really grateful to the reviewer #2 for positive assessment on the revised manuscript. Regarding to the scientific expression suggested by the reviewer, we have carefully read through the Introduction part and corrected according descriptions. Since the reviewer still concerned about the novelty of this study, we add new arguments on the novelty of this study.

Because oxygen has a high electronegativity, its presence can have a marked effect on the properties of materials. This can occur through the binding of free electrons, which leads to a loss of electrical conductivity (e.g., in metal oxides) through the presence of vacancies in the oxygen sublattice. The presence of oxygen atom has a strongly effect on the mechanical properties of nanostructured materials via forming oxides or interstitial atoms, leading the distortion of crystal lattice as very recently shown for carbon interstitial atoms in steel [Ref.1]. Up to now, it is still unclear on the exact role of interstitial atoms behaves in materials, e.g. in nanostructured materials. This is urgently demanded for developing the novel nanostructured materials and for potential applications of such nanocrystalline materials in industry.

The work presented here is a first study on oxygen behavior in nanostructured materials prepared by severe deformation. Our results revealed that in addition to the decomposition process, for the non-equilibrium nanostructured materials, the nanometer-sized oxide clusters could unexpectedly form at specific temperatures. Moreover, our recently obtained new APT (atom probe tomography) experimental data, together with STEM-EELS mappings, can clearly strengthen our novel observations. The main points for novelty of this work:

- i) This is the first atomic-scale observation of the oxidation and decomposition process in bulk nanocrystalline alloys, which shows an unexpected behavior, and the first study on oxygen impurity effects on the structure and properties of bulk nanostructure materials;
- ii) First experiments to demonstrate the thermal stability of such Cu-Fe nanostructured materials, providing the guide for application and study the stability of other relevant nanostructured alloys.
- iii) The finding is of great significance in developing a new route for designing new bulk nanocrystalline alloys with tunable mechanical properties by intentionally introducing different oxygen contents, which can create nano-sized oxide dispersions in the matrix.
- iv) This study will assist in understanding the dynamic behavior of oxygen in other material systems, i.e. oxides and steels, and clarifying the role of oxygen in the field of surface science and catalysis.

Reference

[1]. S. Djaziri, Y. Li, G.A. Nematollahi, B. Grabowski, S. Goto, C. Kirchlechner, A. Kostka, S. Doyle, J. Neugebauer, D. Raabe, G. Dehm, *Deformation-Induced Martensite : A New Paradigm for Exceptional Steels*, **Adv.Mater.**, 28(2016) 7753-7757.

Reply to Reviewer #3:

We thank the reviewer for the positive remarks. Regarding the two further observations, we reply as follows

- 1) As for the judgement of oxides positions via observing moiré fringes, we agree with the reviewer it is hard to prove that the oxides are growing on the surface of the samples. Now we have provided the direct evidence to prove that the oxides are formed interior the grains via atom probe tomography (APT) as showed in Figure 3d and also nucleated at the grain boundary (supplementary Fig 6). The details about this concern can also be referred to the replies to the Reviewer #1 (Question 2 and 4).
- 2) For the two papers recommended by the reviewer, we have already added them in the revised version of manuscript. Thanks for this recommendation.

Reviewers' Comments:

Reviewer #1:

None

Reviewer #3:

Remarks to the Author:

I believe that the addition of new APT results removes some of the previous ambiguities highlighted by the referees. I would therefore now be prepared to recommend publication of the article in its revised version.

Reviewer #4:

Remarks to the Author:

It seems to me that the authors give detailed answers to the Reviewer1's queries. I believe that discussion between Reviewer1 and the authors does not concern with the main idea of the manuscript, but aims to some technical features of TEM experiments. Professor Zhang and his co-authors present the first direct (TEM) evidence that majority of oxide particles is situated inside thin foils of their Cu-Fe samples. Moreover, they maintain their statement by newest experimental data (APT). Naturally, these highest-class experiments with very complicated metallic objects / materials should induce some skepticism in materials science community as it was with the first steps of TEM in 40-50 years of last century.

This work establishes the highest standard of the structural study in materials science of metallic nanomaterials and I would suggest the Editor to accept it for publication. However, I believe that professor Zhang should be the first author of this publication because IT IS NOT A STUDENT'S WORK!

Reply to Review Comments:

We are very grateful to the reviewer #3 and reviewer #4 for positive and valuable comments on our manuscript (NCOMMS-17-04523B-Z). We address the comments raised by reviewers point-by-point as follows:

Reply to Reviewer #3:

We thank the reviewer for constructive suggestions.

1. Point 1 is a semantic argument about whether a whole region of the sample is “perfectly” on a zone axis. This is clearly not the case, for the reasons stated, and appears to have been conceded by the authors.

Thanks. We agree with the reviewer’s explanation, and we have declared in the last response to reviewers’ comments, that it was inappropriate to use “perfectly” to describe the zone axes.

2. Points 2, 3 and 4 relate largely to the location of the oxide particles. Are they formed within the sample, on the grain boundaries, or on the surface? The new APT data plus a STEM-EELS line scan appear to suggest that at least some of the particles are formed within the grains, but the authors need to make clear in the text that this is in addition to any particles forming at grain boundaries, or surfaces, which cannot be precluded.

We agree with the reviewer’s suggestion. We add one sentence at the end of the discussion paragraph. Nevertheless, it should be mentioned that although nano-sized oxide clusters are detected inside grains during heating, the possibility that oxides precipitate at grain boundaries and TEM foil surfaces cannot be fully excluded.

3. I still have reservations about the novelty of this study, but I leave that decision to the editors.

Thanks. About the novelty of this study, we have declared clearly in the last reply.

Reply to Reviewer #4:

We really thank the reviewer for giving such high appraisal to our study.